# Dietary restriction induces posttranscriptional regulation of longevity genes

Jarod A Rollins[1] , Dan Shaffer[1], Santina S Snow[1], Pankaj Kapahi[2], Aric N Rogers[1]

**Dietary restriction (DR) increases life span through adaptive changes in gene expression. To understand more about these changes, we analyzed the transcriptome and translatome of *Caenorhabditis elegans* subjected to DR. Transcription of muscle regulatory and structural genes increased, whereas increased expression of amino acid metabolism and neuropeptide signaling genes was controlled at the level of translation. Evaluation of posttranscriptional regulation identified putative roles for RNA-binding proteins, RNA editing, miRNA, alternative splicing, and nonsense-mediated decay in response to nutrient limitation. Using RNA interference, we discovered several differentially expressed genes that regulate life span. We also found a compensatory role for translational regulation, which offsets dampened expression of a large subset of transcriptionally down-regulated genes. Furthermore, 3′ UTR editing and intron retention increase under DR and correlate with diminished translation, whereas trans-spliced genes are refractory to reduced translation efficiency compared with messages with the native 5′ UTR. Finally, we find that *smg-6* and *smg-7*, which are genes governing selection and turnover of nonsense-mediated decay targets, are required for increased life span under DR.**

## Introduction

Methods of dietary restriction (DR) that increase life span involve restricting nutrients without causing malnutrition. Many forms of DR exist, including restriction of calories, specific macronutrients or micronutrients, and the timing of access to food (Greer and Brunet, 2009; Honjoh et al, 2009,). Along with extended longevity, DR increases resistance to stress and improves other parameters of health from yeast to mammals (Mair and Dillin, 2008), indicating that physiological responses to DR are evolutionary conserved. It is not surprising, then, that much attention has focused on understanding how animals adapt to DR. Identifying genetic programs that control the health benefits of DR guides efforts to develop drug mimetics, which would replace extreme dietary interventions to

increase healthy longevity. To support such efforts, investigators need to understand the genes involved in adapting to DR so they can determine which ones impart benefits associated with this regimen.

Several molecular pathways and cellular processes are important for the effects of DR, especially those involved in nutrient sensing and energy status (Guarente, 2011; Kapahi et al, 2010; Kenyon, 2005). For example, energy sensing via AMP kinase and a greater role for energy production through aerobic respiration, along with changes to DNA and chromatin, are among the adaptive changes in response to DR (Vellai et al, 2003; Apfeld, 2004). Highly conserved pathways shown to have roles in increased life span under DR include the insulin/insulin-like signaling (ILS) and mechanistic target of rapamycin (TOR) pathways. In response to changes in nutrient availability, the ILS pathways modulate cellular processes and coordinate responses in different tissues through hormone signaling (Lin et al, 2001), whereas the TOR pathway can modulate cellular responses by directly sensing nutrients within the cell (Rohde et al, 2001). Changes in transcription associated with these pathways has been pivotal in resolving connections to biological processes and identifying new targets involved in increased life span (Weindruch et al, 2001; Han & Hickey, 2005; Zeier et al, 2011; Palgunow et al, 2012). However, studies have shown that these pathways also direct translation (Kimball et al, 1994; Long et al, 2002), including selective translation of specific mRNA species (Zid et al, 2009; McColl et al, 2010; Rogers et al, 2011; Thoreen et al, 2012).

The need to account for different levels of gene regulation is exemplified in a number of high-profile studies showing that mRNA fluctuations, in general, account for less than half of the variability in corresponding proteins from yeast to humans (Ghaemmaghami et al, 2003; de Godoy et al, 2008; Schwanhäusser et al, 2011; Wilhelm et al, 2014). Thus, a comprehensive understanding of gene expression remodeling that takes place under DR requires an approach that accounts for transcript abundance and the efficiency with which individual transcripts are used to synthesize new protein. This approach provides the basis for development of hypotheses and tests of mechanisms that control gene expression at different levels. For example, it allows for the identification of putative sequence motifs within mRNA (i.e., *cis*-regulatory elements) and corresponding miRNAs and RNA-binding proteins (RBPs) (i.e., *trans*-acting factors) targeting these motifs to ascertain their roles in translational control.

---

[1]Mount Desert Island Biological Laboratory, Salisbury Cove, ME, USA   [2]Buck Institute for Research on Aging, Novato, CA, USA

Correspondence: arogers@mdibl.org; jrollins@mdibl.org

Today, methods involving polysome profiling (Zong et al, 1999) and ribosome footprinting (Ingolia et al, 2009) provide information that can be used to help distinguish forms of posttranscriptional regulation. By isolating mRNA that is associated with actively translating ribosomes, polysome profiling allows the translation state of individual transcripts to be quantified between treatments. In comparison, ribosome footprinting allows for the distribution of the ribosome along mRNA to be quantified by sequencing only the regions of transcripts protected from nuclease digestion by bound ribosomes. These technologies have been applied in yeast to characterize the role of 5′ UTRs in transcript stability and translatability (Ringnér & Krogh, 2005), to pinpoint exonic polymorphisms in mRNA that affect translation, and to analyze stress-induced changes in translation of specific transcripts (Lackner et al, 2012). Polysome profiling and ribosome footprinting have also been used to consider isoform-specific translational regulation (Sterne-Weiler et al, 2013) and to assess selective mRNA translation during mechanistic TOR inhibition in cell culture (Olshen et al, 2013). These studies and others like them have started to resolve the importance of posttranscriptional mechanisms in regulating gene expression. However, there is still a paucity of information on the relationships that exist between genes controlled transcriptionally, translationally, or both in mediating the adaptive response to DR. Understanding these relationships is key to understanding how DR promotes healthy aging.

In the following study, *Caenorhabditis elegans* adult nematodes were subjected to a form of DR involving food dilution that results in robust life span increase (Chen et al, 2009). Subsequently, animals were assessed for gene expression changes according to total and polysome-associated mRNA to differentiate transcriptional and posttranscriptional inputs. mRNAs differentially associated with polysomes under DR were investigated in silico for characteristics that could explain changes in gene expression at this level. Genes transcriptionally and/or translationally down-regulated under DR were screened using RNAi for effects on life span under fully fed conditions. Results reveal novel genes and connections with forms of posttranscriptional regulation important for increased life span under DR.

## Results

### Transcriptional responses to DR implicate muscle and metabolic adaptation

Using food dilution (Chen et al, 2009) to impose DR starting at day 1 of adulthood extended median life span of *C. elegans* by ~40% (Fig 1A). To better understand the different inputs to gene expression, we analyzed total and translated (i.e., polysome-associated) mRNA after 4 d under DR or fully fed (ad libitum, or AL) conditions in adult nematodes. Using four biological replicates, total and translated mRNA from AL and DR worms were analyzed by next-generation sequencing (Fig 1B). In brief, 100 bp paired end reads were aligned to the *C. elegans* genome and reads were counted using HTseq (Fig 1C). edgeR was used to normalize the dataset and filter out lowly expressed mRNA that could not be reliably quantified (see the

Materials and Methods section), leaving 8,301 annotated coding genes in the analysis of differential expression. A heatmap representing expression across this dataset demonstrated that the biological replicates clustered together, that the expression dataset was normally distributed, and that several instances of differential expression existed between different groups (Fig S1A).

To compare total mRNA expression between AL and DR conditions, we identified genes that were significantly changed twofold or more between diets (false discovery rate [FDR] < 0.05; Fig 1D). Of these, 257 were up-regulated under DR and 782 were down-regulated (Table S1). Gene ontological (GO) analysis was then used to determine whether particular biological processes were represented among differentially expressed genes. Up-regulated genes were enriched for processes related to myosin and myofibril assembly, muscle contraction, locomotion, oxidation/reduction, and ion transport (Fig 1E and Table S2), whereas down-regulated genes were enriched for biological processes related to apoptosis, amino acid metabolism, defense response, epigenetic regulation, RNA metabolism, and transcription (Fig 1F and Table S2).

Among the up-regulated genes, expressions of several genes are known to be critical for age-related health and longevity. Like age-related sarcopenia in humans, locomotion and the ability to transduce force generated by muscles is lost with age in *C. elegans* (Hahm et al, 2015; Rollins et al, 2017). Under DR, the genes encoding thick filaments (*unc-54* and *myo-1*) and movement-coordinating troponins (*pat-10*, *tni-3*, and *tnt-4*) of muscle sarcomeres were up-regulated (Table S2), suggesting that they may be involved in the maintenance of locomotion with age in DR restricted animals. In line with this, *C. elegans* mutants with defective *unc-54* activity have reduced capacity to generate movement force (Várkuti et al, 2012). In addition, overexpression of *pat-10* leads to increased actin filament stability, resistance to protein unfolding stress, and life span (Baird et al, 2014). Among the genes with roles in oxidation reduction were members of the cytochrome P450 family (*cyp-34A9*, *cyp-35A2*, *cyp-33C3*, and *cyp-34A4*). Cytochrome P450 isozymes can serve in the metabolism of xenobiotics and other toxins (Ding & Kaminsky, 2003; Lindblom & Dodd, 2006), which might otherwise impinge on longevity.

Defense response genes were down-regulated, including those encoding neuropeptide antimicrobials (e.g., *npl-29* and *npl-31*) in addition to *tol-1*, the only Toll-like receptor in *C. elegans*. Although efficacy of defense responses is usually improved in DR animals, diminished expression of these genes may be because of reduced microbe sensing resulting from the form of DR itself, which involves dilution of their bacterial food source. The transcription factor GO term included regulators of embryonic development, which were expected to be decreased given the negative effect of DR on reproduction. These included nuclear hormone receptors (e.g., *nhr-2* and *nhr-112*), cyclin-dependent kinase 8 (*cdk-8*), a homeobox factor (*ceh-40*), and a HAND bHLH factor important for tissue differentiation (*hnd-1*), among others (Table S2). The heat shock transcription factor gene *hsf-1* was also down-regulated, presumably because of globally reduced protein synthesis, which requires *hsf-1*–dependent chaperones to direct the folding of nascent proteins.

In a previous study, transcript levels were analyzed in *C. elegans* subjected to DR based on intermittent fasting (Honjoh et al, 2009). We compared our list of differentially regulated genes from bacterial

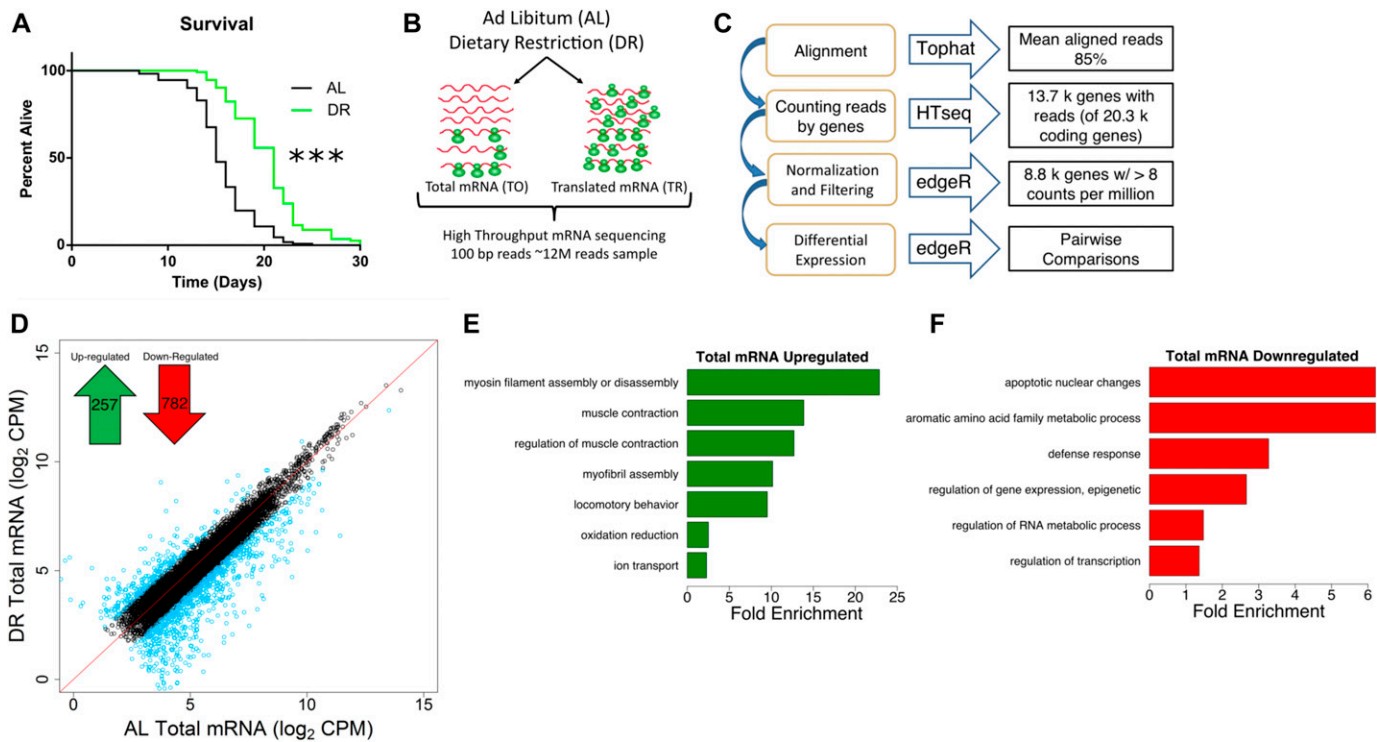

**Figure 1. Transcriptional mRNA regulation under DR in *C. elegans*.**
**(A)** DR increased median life span by 40% compared with well-fed conditions. *P* < 0.0001, n = 120, log-rank test. **(B)** Schematic representation of experimental setup combining mRNA-seq and polysome profiling to quantify gene expression. At day 1 of adulthood, nematodes were placed on AL or DR plates for 4 d. Total (TO) and translated (TR) mRNA were extracted from whole worm populations in biological quadruplicate and subjected to high-throughput sequencing. **(C)** Schematic representation of deep sequencing analysis. Reads were aligned to the *C. elegans* reference genome WS220 using Tophat and were counted using HTseq. Normalization, filtration, and differential expression were performed by edgeR. A total of 8,301 genes were quantified. **(D)** Protein-coding genes are differentially expressed under DR on the transcriptional level as determined using reads from total mRNA. Each circle represents expression for a specific gene. Black circles plotted near the red line represent genes with the same or similar expression between diets, whereas blue circles represent genes with twofold or greater change. Units are $\log_2$ values of counts aligned per million reads sequenced (CPM). Inset, 257 protein-coding genes were up-regulated and 782 genes were down-regulated significantly (FDR < 0.05) twofold or more under DR. **(E, F)** Biological GO terms significantly (*P* < 0.05, modified Fisher's exact test) enriched among genes transcriptionally up-regulated or down-regulated under DR, respectively.

dilution with those obtained by Honjoh et al (2009) using gene set enrichment analysis (GSEA, Fig S1B). This analysis demonstrated a significant (FDR < 0.001) enrichment of both up- and down-regulated genes between the two datasets despite differences in dietary regimes and quantification methods (i.e., next-generation sequencing versus microarray). To gain insight into the transcriptional differences between DR by food dilution and intermittent fasting, differentially expressed genes in common between DR methods and those unique to food dilution were subjected to GO analysis. Terms related to oxidation/reduction and muscle contraction were up-regulated in common among both forms of DR (Table S2). Among down-regulated genes, terms in common pertained to defense and immune response. However, the term "regulation of transcription" was unique to down-regulated genes in the food dilution DR dataset (Table S2). Together, results demonstrate that many of the same processes regulated transcriptionally by intermittent fasting are also regulated by food dilution in *C. elegans*.

To gain a better understanding of which responses to DR are species specific and which are conserved among different species, we compared our differential expression results with a meta-analysis performed by Han and Hickey (2005). In the meta-

analysis, they compared differently expressed genes in mouse, rat, pig, monkey, yeast, and fly subjected to DR. Interestingly, their study showed a lack of overlap in expression changes of specific genes in common to all these species. However, GO analysis did point to the conservation of biological processes involved in adaptive responses to DR like those observed in the present study. Specifically, the meta-analysis reported the enrichment of genes annotated with the term metabolism among all animals listed. In addition, there was a high level of overlap for terms related to stress/immune response and regulation of transcription that were the same or similar in *C. elegans* (Fig 1F). These similarities imply evolutionary conservation of transcriptional responses to DR at the level of biological processes.

### Translation frequently offsets transcriptional down-regulation under DR

To differentiate transcriptional and translational regulation of gene expression, we analyzed the relative abundance of the 8,301 quantified genes within the total and polysome-associated mRNA pools. Under AL conditions, 82% of genes had similar abundance

between pools. The remaining 18% showed a twofold or greater difference between pools and were considered to be translationally regulated (Fig 2A). Thus, genes with greater relative abundance among polysomes than total mRNA were considered to be translationally promoted, whereas those with reduced polysomal

association were considered translationally suppressed. In contrast to AL conditions, 34% of genes were translationally regulated under DR (Fig 2B). This is reflected by reduced correlation between total and translated mRNA pools (r = 0.88 for AL versus 0.83 for DR, P < 0.0001, t test of Fisher r-z transformed coefficients). In addition,

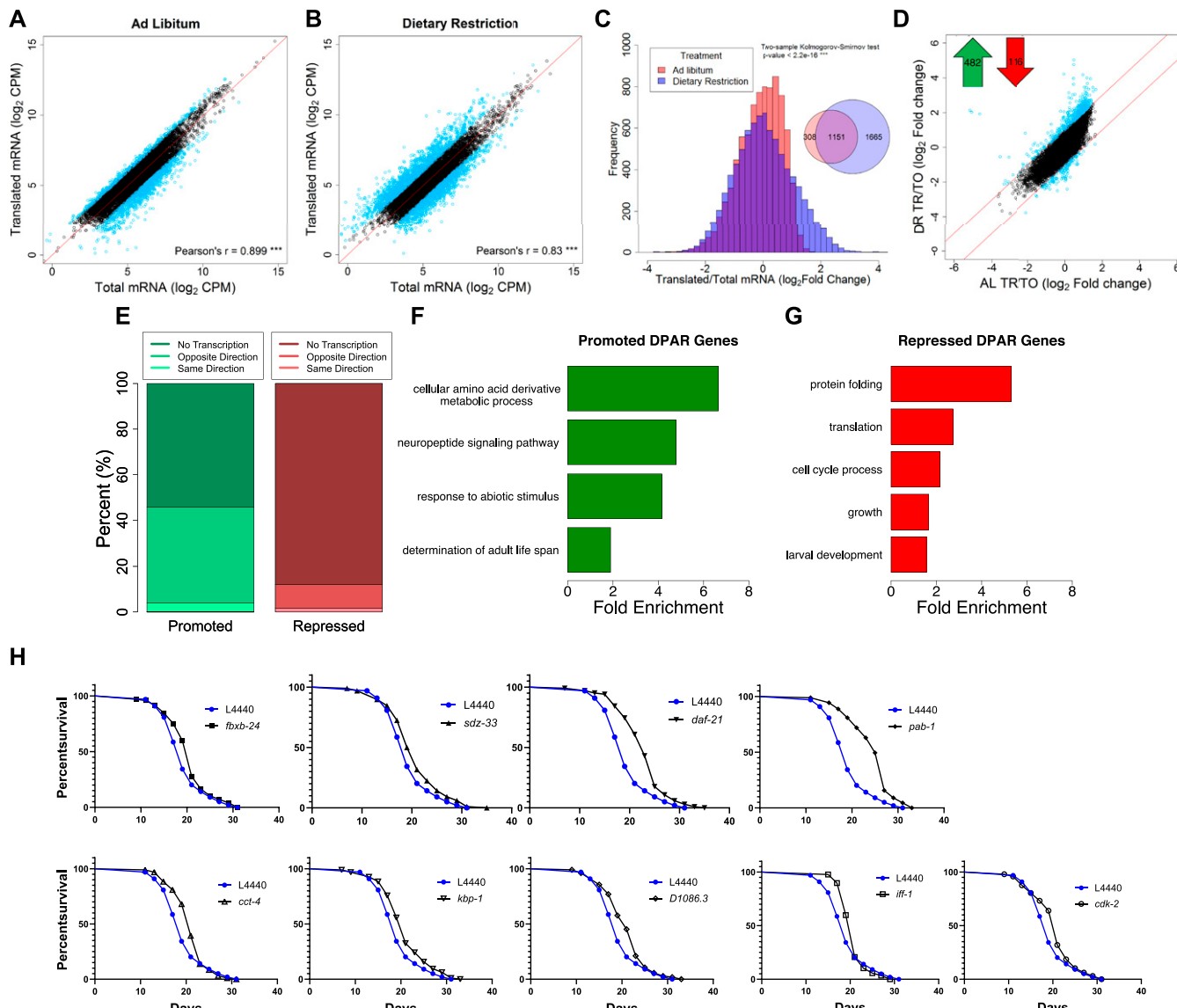

**Figure 2. DR modulates selective translation in *C. elegans*.**
**(A, B)** Gene abundance between fractions was well correlated under (A) AL and (B) DR (*P*-value < 0.001, Pearson's correlation of 0.899 and 0.830, respectively). Black circles represent genes that showed no significant difference (FDR > 0.05) between total and translated mRNA. Blue circles represent genes with twofold or greater change. A reference line with a slope of one is shown in red. Units are counts aligned per million sequenced (CPM). **(C)** Distribution of gene abundance fold changes between translated and total mRNA for AL and DR (blue) worms. The distribution of changes was significantly different between diets (two-sample Kolmogorov–Smirnov test, *P* < 2.2 × 10⁻¹⁶). **(D)** Translational promotion and repression of genes between diets were calculated using the DPAR. Black circles represent genes with similar translation: transcription expression ratios between diets. Blue circles represent genes with twofold or greater DPAR. Red lines delineate the twofold change cutoff. Inset, 482 genes were translationally promoted and 116 genes were translationally repressed. **(E)** Percent of translationally promoted or repressed DPAR genes in which changes in the total fraction were either in the same direction (light hue), opposite direction (medium hue), or were not significant (dark hue; Log2 FC > 2, FDR < 0.05). **(F, G)** Biological GO terms associated with translationally (F) promoted or (G) repressed DPAR genes under DR. *P* < 0.05, modified Fisher's exact test. Functionally similar GO terms from identical groups of genes were manually removed. **(H)** Genes down-regulated under DR which resulted in life span extension when knockdowned in *C. elegans* under well-fed conditions. One representative replicate is shown; all replicates performed can be found in Table S6. The % change in median life span and *P*-value from the log-rank test comparing with L4440 empty vector controls are: *fxbx-24* (10.5%, 0.030), *sdz-33* (5.3%, 0.025), *daf-21* (21.1%, 2.00E-16), *pab-1* (31.6,% 2.00E-16), *cct-4* (10.5%, 0.011), *kbp-1* (10.5%, 0.006), *D1086.3* (10.5%, 0.005), *iff-1* (10.5%, 0.004), *cdk-2* (10.5%, 0.031). fxbx-24 and sdz-33 were regulated transcriptionally under DR, D1086.3 was regulated transcriptionally and translationally, and all other genes were predominantly regulated translationally.

the distribution of changes in gene expression between the total and translated fractions was significantly different under DR compared with AL ($P < 2.2 \times 10^{-16}$, two-sample Kolmogorov–Smirnov test, Fig 2C). As seen in the Venn diagram inset in Fig 2C, the majority of selectively translated genes under AL were also subject to translational regulation under DR. Thus, a common set of genes are translationally regulated under both conditions, whereas an additional set of genes becomes translationally regulated under DR.

To characterize the predominant modes of gene regulation that occur under DR without bias, we clustered genes that only become translationally regulated under DR (1,665 in all, see Venn diagram in Fig 2C) into groups with similar expression profiles. Partitioning around medoids was used to cluster genes into the optimal number of groups using the gap statistic (Tibshirani et al, 2001). These genes were optimally clustered into three groups as seen in a principle compoment analysis plot that explained 97.6% of the dataset variance (Fig S2A). The relative expressions of all 1,665 genes for total and translated mRNA under AL and DR are shown for each of the three clusters in Fig S2B–D. The first cluster consisted of 530 genes exhibiting down-regulated transcript levels under DR that were partly offset by positive selective translation (Fig S2B). The second group contained 492 genes lacking transcriptional changes with modest negative translation changes under AL that became more negative under DR (Fig S2C). The third cluster contained 643 genes that were essentially unchanged transcriptionally under both conditions but that become translationally promoted under DR (Fig S2D). Although other modes of regulation exist under DR, these clusters account for the majority of expression changes. Thus, it appears that there are three fundamental roles of translational regulation that occur under this DR regimen: to oppose transcriptional down regulation (cluster 1), to reinforce translational suppression tendency observed under AL (cluster 2), and to increase expression during DR in the absence of transcriptional changes (cluster 3).

The analysis in Fig S2 helps resolve patterns of expression for genes that only become translationally regulated subsequent to DR. To determine which genes exhibit net translation changes between conditions, we devised the Differential Polysome Association Ratio (DPAR). This is calculated by taking the ratio of translated to total mRNA under DR and dividing by the same ratio under AL for a given gene. It should be noted that, while global translation rates are lower under DR, DPAR is only influenced by relative changes in the distribution of specific mRNAs within polysomes and total mRNA. A significant (FDR < 0.5) twofold or greater change in translation was observed for 482 translationally promoted genes and 116 translationally suppressed genes (Fig 2D and Table S3). A threshold of twofold change in polysome association was used to focus the analysis on genes experiencing large changes in translational regulation and comes with the tradeoff of potentially missing smaller but still biologically relevant regulation. Interestingly, the tendency for translationally changed genes to be up-regulated under DR was opposite of the tendency for transcriptionally changed genes, which were much more frequently down-regulated (compare Figs 1D and 2D). This led us to investigate how often translation offsets transcriptional regulation under DR. Of the genes that were translationally promoted under DR, 58% lacked significant transcript level changes and 40% had significantly decreased transcript level changes, leaving very few genes

that were both transcriptionally and translationally promoted (Fig 2E). Given that only 12.5% of quantifiable genes in our dataset displayed significant transcriptional regulation, the fact that 33% of translationally promoted genes were significantly down-regulated at the transcript level was, itself, significant ($P < 2.2 \times 10^{-16}$, Fisher's exact test). Conversely, 90% of translationally down-regulated genes showed no change in transcription. The frequent opposition between translation and transcription may help explain numerous studies showing disparate results between gene transcription and protein expression (Ghaemmaghami et al, 2003; Schwanhäusser et al, 2011; Wilhelm et al, 2014).

## Translational adaptation to DR involves down-regulating anabolic processes and altering amino acid metabolism and neuropeptide signaling

We performed GO enrichment analysis of DPAR genes to determine which biological processes are candidates for DR-relevant adaptation through translation. Translationally promoted DPAR genes were enriched ($P > 0.05$) for processes related to amino acid metabolism, neuropeptide signaling, abiotic responses, and determination of adult life span (Fig 2F and Table S4). One example is gst-10, a glutathione S-transferase for which overexpression has been shown to increase median longevity in C. elegans by 22% (Ayyadevara et al, 2005). Interestingly, translationally promoted DPAR genes dod-19, dod-23, and mtl-2 were previously shown to be down-regulated in daf-2 ILS mutants and could further increase life span when knocked down in the daf-2(mu150) background (Murphy et al, 2003). In the case of dod-19 and dod-23, their transcriptional down-regulation under DR (>24-fold and >5-fold reduction, respectively) was far greater than their translational promotion (eightfold and twofold increase, respectively), which is likely to result in an overall reduction in new protein synthesis of these genes under DR, consistent with their negative roles in longevity. The mtl-2 mutant has a life span similar to wild-type worms (Hughes & Stürzenbaum, 2007), which is in line with a context-dependent role in longevity. Among the translationally suppressed genes, there was an enrichment of biological terms related to protein folding, translation, cell cycle, growth, and development (Fig 2G and Table S4). These results are in agreement with effects of DR on limiting growth and decreasing global translation.

## RNAi screen of translationally regulated genes under DR revealed novel longevity genes

As DR prolongs longevity, we tested whether inhibiting genes that are down-regulated under food dilution increases life span in C. elegans. To avoid deleterious effects on growth, RNAi was initiated after development was complete. We included the top 100 transcriptionally down-regulated genes and the top 95 translationally down-regulated genes under DR using the Ahringer (Fraser et al, 2000; Kamath et al, 2003) and ORFeome (Rual et al, 2004) RNAi libraries. The genes tested are listed in Table S5.

Three of the 100 transcriptionally suppressed genes resulted in a significant extension in median life span when targeted with RNAi: fbxb-24, sdz-33, and D1086.3 (Fig 2H and Table S6). The genes fbxb-24 and sdz-33 encode proteins containing F-boxes, which have been

implicated in mediating protein–protein interactions important for cell cycle regulation and signal transduction (Craig & Tyers, 1999). Little information is available for D1086.3, but it is the only gene both transcriptionally and translationally down-regulated with a life span phenotype (Table S6 and Fig 2H). The list of screened genes also included *dod-19*, *dod-23*, and *clec-186*, which have been found in previous studies to increase life span when knocked down but which did not increase life span in the current study. *dod* genes are named for being downstream of DAF-16, a forkhead box O transcription factor required for increased life span when the ILS pathway is attenuated (Murphy et al, 2003). These *dod* genes are down-regulated when the ILS pathway is genetically restricted, and targeting them with RNAi in *daf-2(mu150)* animals further increases the long life span associated with background (Murphy et al, 2003). Our results indicate that these *dod* genes may require muted ILS to effectively increase life span when they are targeted with RNAi. Knockdown of the innate immunity gene *clec-186* was shown to extend life span in *C. elegans* with increased sensitivity to RNAi (Hamilton et al, 2005), suggesting that the total or tissue-specific extent of suppression is a factor for its role in life span.

Seven of the 95 down-regulated DPAR genes increased life span when targeted for knockdown via RNAi, including D1086.3. The other six genes are *daf-21*, *cct-4*, *cdk-2*, *iff-1*, *pab-1*, and *kbp-1* (Fig 2H). The *daf-21* gene encodes the ortholog of the mammalian molecular chaperone HSP90. Despite its title as a heat shock protein, HSP90 is a pro-growth chaperone that inhibits activity of the heat shock transcription factor (Kijima et al, 2018). Lowering its expression was recently shown to increase life span in wild-type animals in a separate study investigating *C. elegans* orthologs of human genes that are differential expressed with age in blood (Sutphin et al, 2017). *cct-4* encodes for a chaperonin required for SKN-1–dependent transcription of a putative glutathione-requiring prostaglandin D synthase, *gst-4* (Kahn et al, 2008). The gene *cdk-2* encodes for a cyclin-dependent kinase, which has been recently implicated in regulating stress responses and longevity by inhibiting cell cycle machinery (Dottermusch et al, 2016). A germ line–specific translation initiation factor is encoded by *iff-1*, which was shown to regulate life span in the first genome-wide functional screen for longevity genes (Hamilton et al, 2005). Another translation initiation factor, *pab-1* encodes a protein that binds the poly-A tail of mRNA and is discussed more in the next section. The gene *kbp-1* is involved in the cell cycle by regulating kinetochore–microtubule binding (DeLuca & Salmon, 2004) and is necessary for larval development (Rual et al, 2004). These life spans were conducted an additional time without 5'-fluorodeoxyuridine (FuDR) to examine if it impacted the longevity effect of the RNAi treatments (Table S6). Results show comparable effects to the replicates conducted with FuDR. To our knowledge, our findings for *kbp-1* and D1086.3 were not previously shown to play a role in regulating aging in *C. elegans*. Furthermore, these results demonstrate the importance of understanding the role of translational regulation of gene expression in longevity.

### Translationally suppressed genes under DR are frequently in operons

Analysis of the switch from AL to DR showed that translationally promoted genes were frequently associated with diminished

transcript levels (Figs 2E and S2B), pointing to a possible link between transcription and translation in dietary adaptation. To address mechanisms governing translation, and a possible connection to transcriptional regulation, we investigated the roles of operons and *trans*-splicing. The genome of *C. elegans* contains operons consisting of two or more genes that are transcribed together but separated posttranscriptionally and translated individually (Zorio et al, 1994). We hypothesized that expression of genes of the same operon may require compensatory translational regulation in response to changes in nutrient availability. Using previously established operon annotations (Allen et al, 2011), we calculated the percent of genes in operons among all coding genes and compared this with the percent found among translationally promoted or suppressed DPAR genes (Fig 3A). Of the 482 promoted genes, a significantly smaller proportion (10%, $P < 0.001$, Fisher's exact test) were organized in operons compared with 8,301 coding genes (18%) with quantifiable expression in our dataset. Conversely, the genes translationally suppressed contained a higher proportion of operonic genes (50%, $P < 0.05$, Fisher's exact test). This observed enrichment suggested that the translational suppression observed under DR may serve to suppress the expression of one or more genes that are co-transcribed at levels higher than needed under DR. This pattern was inverted among the majority of translationally promoted genes, which tended not to be associated with operons (Fig 3A) and which, instead, tended to be associated with transcriptionally suppressed genes (Fig 2E). To gain a better understanding of this association, we addressed the known attributes of operonic and nonoperonic genes that can influence their rate of translation.

### *Trans*-spliced genes are refractory to diminished translation under DR compared to other genes

The 5′ UTR of many *C. elegans* mRNA are *trans*-spliced with one of a small number of short spliced leader (SL) sequences that replace much of the originally transcribed 5′ UTR. 5′ UTRs are important for regulating translational efficiency, and *trans*-splicing has a positive effect on translation in unstressed *C. elegans* (Yang et al, 2017). Commonly, the first gene in an *C. elegans* operon is *trans*-spliced to SL1, and all downstream transcripts are *trans*-spliced to SL2 (Allen et al, 2011). For genes that are not in operons, 5′ UTRs are frequently *trans*-spliced to SL1, with around 30% of transcripts retaining the native 5′ UTR (Allen et al, 2011).

Using annotations of SL1 and SL2 spliced genes from Allen et al (2011), we calculated the average fold change in translational regulation of genes with *trans*-spliced or native 5′ UTRs. Under AL, our analysis confirmed that both SL1 and SL2 *trans*-spliced messages tend to be better translated than transcripts with native 5′ UTRs (Fig 3B), as previously reported (Yang et al, 2017). DR led to translational suppression of SL1 spliced genes compared with AL conditions ($P = 1.4 × 10^{-7}$, Wilcoxon rank sum test with continuity correction). In contrast, the translation efficiency of SL2 spliced genes was unchanged under DR ($P = 0.11$, Wilcoxon rank sum test with continuity correction). For both SL1 and SL2 genes, translation was preserved better in the transition from AL to DR compared with those annotated with native 5′ UTRs.

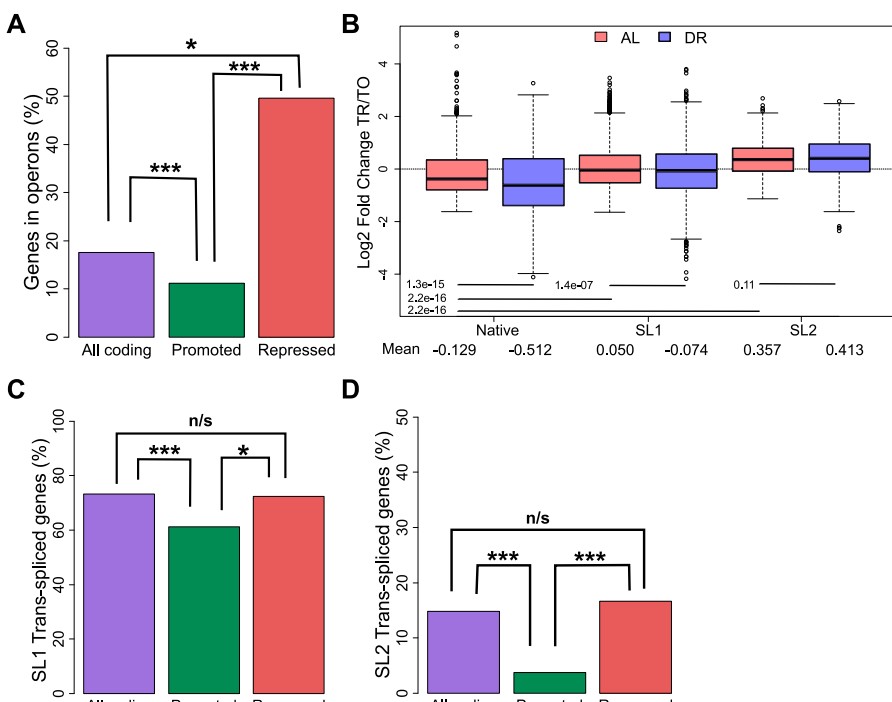

**Figure 3. Operonic genes are enriched among genes that undergo translational repression in the switch to DR.**

**(A)** Percent of operonic genes among all protein-coding genes (purple), translationally promoted DPAR genes (green), or translationally repressed genes (red). **(B)** Translational regulation of genes under AL (light red) or DR (blue). Genes are grouped by their predominant 5′ UTR, which may be native or *trans*-spliced to SL1 or SL2. The mean Log2 fold change between translated and total fractions is given for AL and DR. Group mean values were compared using Wilcoxon rank sum test and the associated *P*-values are given. **(C)** Percent of genes SL1 trans-spliced among all protein-coding genes (purple), translationally promoted genes (green), or among translationally repressed genes (red). **(D)** Percent of SL2 spliced genes among all protein-coding genes, translationally promoted genes, and translationally repressed genes under DR. TO, total fraction; TR, translated fraction. Asterisks denote significance (*P < 0.05, ***P < 0.001) using Fisher's exact test.

Approximately 73% of coding genes quantified in this study were annotated as SL1 *trans*-spliced (Fig 3C). In comparison, a significantly lower proportion of SL1 *trans*-spliced transcripts were among promoted DPAR genes (61%, *P* < 0.01, Fisher's exact test), whereas a similar proportion was observed among repressed genes (63%, *P* > 0.05, Fisher's exact test). A similar analysis for SL2 *trans*-spliced transcripts showed a significantly lower proportion among promoted DPAR genes (4%, Fig 3D and *P* < 0.001, Fisher's exact test) but not among suppressed DPAR genes (17%, *P* > 0.05, Fisher's exact test). Thus, while promoted DPAR genes were depleted among SL1 and SL2 transcripts, the depletion is modest and no change from the average distribution of trans-splicing was observed for repressed DPAR genes, suggesting that *trans*-splicing is unlikely to be a primary driver of differential translation among DPAR genes.

### DR increases 3′ UTR editing, which is associated with reduced translation

Within the RNA-seq dataset we found a number of putative adenosine RNA editing events with the potential to alter translational regulation. The modification of adenosine to inosine (A-to-I) is the most prevalent type of RNA editing in many species and is catalyzed by action of adenosine deaminases acting on RNA. Inosine is read as a guanosine (G) by cellular enzymes, which has the potential to change protein-coding regions and protein-binding sites. The A-to-I modification may also change the secondary structure of noncoding regions. For some genes, like the glutamate receptor in mice, RNA editing is required to produce a functional protein (Wright & Vissel, 2012).

To determine the influence of DR on editing, we quantified A to I conversions occurring within each of four biological replicates for all test groups in our dataset. For an edit site to be included, we required a minimum editing frequency of 10%. Under AL, we detected an average of 125 putative editing sites in the total fraction, which was significantly decreased to an average of 70 in the translated fraction (Fig 4A), indicating that the occurrence of A-to-I editing is negatively associated with translation (*P* = 0.029, Wilcoxon rank sum test with continuity correction). DR amplified this difference, increasing putative editing sites to 154 with a concomitant decrease to 47 sites in the translated fraction (*P* = 0.021, Wilcoxon rank sum test with continuity correction). A similar pattern is observed with respect to the frequency of editing at these sites (Fig S3A). The overlap of editing sites among the different experimental groups is shown in Fig 4B, which shows 39 sites common to all samples, but none that are exclusive to the translated mRNA under DR. A list of the editing sites, their average editing frequencies among the different experimental groups, and the genes in which they occur in are provided in Table S7. The GO term enrichment for biological processes of genes with altered editing frequency is also provided in this table and includes functions pertaining to the proteasome, protein modification, and cell cycle. Together, these results suggest that A-to-I editing may be a method to effectively diminish expression of specific genes under DR.

The composition of 3′ UTRs can influence both transcript stability and translatability (Mayr & Bartel, 2009; Theil et al, 2018; Tushev et al, 2018). Considering all editing sites detected in the dataset, 65% occurred in the coding sequence (CDS), whereas 35% occurred in the 3′ UTR (Fig 4C). The average 3′ UTR length of coding genes quantified in this dataset was only 14.6% of average CDS length, indicating that RNA editing is enriched within this region. To address whether editing of 3′ UTRs influences translation rates, linear regression was performed for CDS-edited and 3′ UTR-edited genes

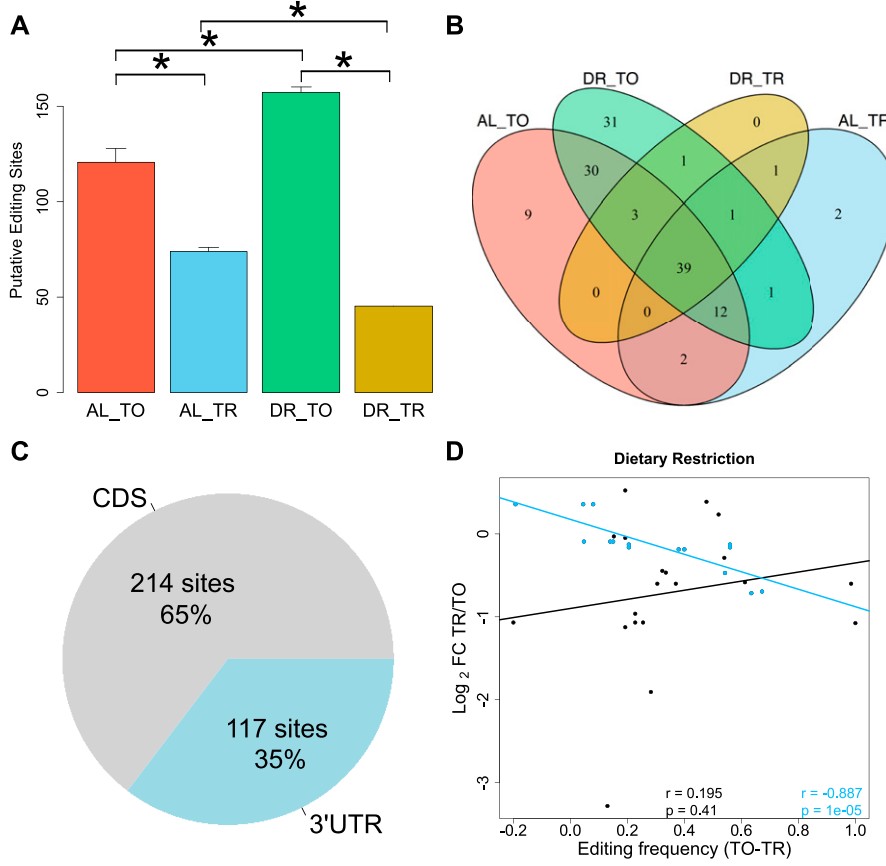

**Figure 4.  mRNA adenosine-to-inosine editing increases under DR.**
**(A)** Average number of RNA edits in total and translated fractions across diets. RNA editing under DR is globally increased in the total fraction yet diminished in the translated fraction as compared with well-fed controls. Error bars are SEM. *$P$ < 0.05, Wilcoxon rank sum test, n = 4. **(B)** Overlap of editing sites present in at least three of four biological replicates in total and translated mRNA under AL or DR. **(C)** Percentage of editing sites within the dataset located in 3′ UTRs as annotated in UTRome and the CDS. **(D)** Relationship between the change in editing frequency and polysome association under DR. Genes with edits occurring in their 3′ UTR are blue, edits occurring in the CDS are black. Regression line for 3′ UTR edits is blue, and line for CDS edits is black. Spearman correlation coefficient and associated $P$-value for 3′ UTR edits and for CDS genes are given in blue and black, respectively. TO, total mRNA; TR, translated mRNA; FC, fold change.

with respect to translational regulation (Fig 4D). Genes with a higher frequency of editing in their 3′ UTRs under DR were more likely to be translationally suppressed ($r = -0.887$, $P = 1 \times 10^{-5}$). No significant correlation was observed for genes CDS-edited messages ($r = 0.195$, $P = 0.41$). Thus, editing events localized to the 3′ UTR negatively correlate with translation under DR.

Among the genes with differential A-to-I editing in their 3′ UTRs was *eif2-alpha*, a subunit of the eukaryotic translation initiation factor-2. This factor is required for delivering charged tRNA$^{Met}$ to the ribosome to initiate translation (Sokabe et al, 2012). Editing of the 3′ UTR of *eif2-alpha* was observed in a comparison of wild-type worms to *adr-1* adenosine deaminase mutants (Washburn et al, 2014). Two edits were present in the 3′ UTR of *eif2-alpha* with significantly different frequencies between total and translated RNA under DR (Fig S3B and C). In accordance with results for mean editing frequency in Fig S3A, the editing frequency at both sites was lower in the translated fraction under DR, suggesting that transcripts edited at these loci are translationally suppressed. The expression of the A-to-I editing regulator gene *adr-1* and the gene encoding its binding partner *adbp-1* was reduced in the translated fraction under DR (Fig S3D). Conversely, the expression of *adr-2* was slightly increased under DR. These expression changes in adenosine deaminases acting on RNA are a likely explanation of the changes in editing frequencies observed under DR. In line with the above observations, the robust translational promotion of *eif2-alpha* under AL conditions was partially attenuated under DR (DPAR of

−0.295, Fig S3E), suggesting that edits in it's 3′ UTR may be responsible for its translational attenuation. The edit occurring at position 2076705 occurs in a stem loop in the predicted secondary structure of the 3′ UTR of *eif2-alpha* (Fig S3F). As stem loops allow for specific recognition of RNA by binding proteins (Mattaj, 1993), this edit could control which regulatory factors bind to the 3′ UTR of *eif2-alpha*. In addition, the edit at position 2076770 removed one of the seed match pairings of mir-1820 and of mir-792 (Fig S3F), which could change the translational regulation of *eif2-alpha*.

## Annotated RNA-binding sequence motifs are enriched among differentially translated mRNA under DR

RBPs and miRNAs can alter the propensity of a target transcript to be translated. Although they differ in their mode of action, both RBPs and miRNAs rely on sequence motifs to recognize mRNA targets. In addition to *cis*-regulatory elements within UTRs of mRNA and corresponding *trans*-acting factors, other characteristics including length and secondary structure can affect translatability (Ringnér & Krogh, 2005; Zid et al, 2009; Rogers et al, 2011). Therefore, we characterized the 3′ UTRs of promoted and suppressed DPAR genes in terms of length, GC content, free folding energy, and presence of RBP or miRNA motifs to determine their relevance in gene regulation under DR.

The average 3′ UTR length of promoted (148.2 bp) DPAR genes were significantly shorter than that of the quantified gene set (180.6

bp; $P = 8.71 \times 10^{-9}$, Wilcoxon rank sum test with continuity correction). No significant differences in mean GC content of translationally promoted or repressed 3′ UTRs were observed. The mean minimal folding energy of promoted 3′ UTRs (−19.1 kcal/mol) were significantly lower than that for the total gene set (−21.5 kcal/mol; $7.19 \times 10^{-05}$, two-sample Wilcoxon rank sum test with continuity correction). In summary, genes translationally promoted under DR tended to have shorter 3′ UTRs with less secondary structure, whereas repressed genes had no distinguishing features in this region (Table 1).

To search for sites of potential interaction with RBPs, we used experimentally determined and/or highly conserved motifs for *C. elegans* maintained at the Catalog of Inferred Sequence Binding Preferences of RBPs (CISBP-RNA) database (Ray et al, 2013). Motif enrichment was calculated separately for 3′ UTRs of translationally promoted or suppressed genes relative to the frequency with which motifs occur among all annotated 3′ UTRs in *C. elegans*. Eleven RBP motifs were significantly enriched ($P < 0.05$) in either one or both sets of DPAR genes (Fig 5A). The motifs for TIAR-1/2, TIAR-3, and EXC-7 were enriched among both promoted and suppressed genes. Target motifs for MEX-5/6, UNC-75, SUP-12, SUP-26, ETR-1, SUP-49, and ASD-2 were enriched almost exclusively among promoted DPAR genes, whereas the target motif for PAB-1 was enriched exclusively among translationally suppressed genes.

Of these RBPs, MEX-5/6, SUP-26, and PAB-1 were reported to influence translation of mRNA in *C. elegans*. MEX-5/6 interferes with other RBPs in early differentiation stages during development (Oldenbroek et al, 2013). SUP-26 binds the sex determination gene *tra-2* in a 28-nucleotide repeat element located in its 3′ UTR to lower protein expression, most likely by inhibiting translation (Mapes et al, 2010). *pab-1* encodes poly-A–binding protein, which recent evidence suggests is important for mediating repression and

deadenylation through interactions with miRNAs targeting the same transcript (Flamand et al, 2016). However, it is primarily considered a pro-translation factor (Burgess & Gray, 2010; Smith et al, 2014). Each of these RBPs were, themselves, down-regulated translationally under DR (Fig 5B). Unlike MEX-5/6 and SUP-26, PAB-1 targets were enriched among translationally repressed DPAR genes. We tested the requirement of these genes for longevity- and DR-induced longevity using RNAi. Knockdown of *mex-5* had no significant effect on longevity under AL or DR (Fig 5C left, Table S8). Knockdown of *sup-26* under AL increased median life span by 26% (Fig 5C right, Table S8). However, knockdown of *sup-26* under DR did not significantly extend life span compared with controls on DR. Under AL, knockdown of *pab-1* extended median life span by 36% (Fig 5C middle, Table S8) in agreement with previous reports with its effect on longevity (Reis-Rodrigues et al, 2012). Under DR, knockdown of *pab-1* was not longer lived than DR controls. Our results for *sup-26* and *pab-1* suggested they are epistatic to DR, as their knockdown under DR did not result in an additional increase in life span. As *pab-1* encodes for poly-A–binding protein, a factor that increases translation of mRNA to which it is bound, its down-regulation under DR may be pro-longevity as a general effect of widespread decreased translation.

The genes *unc-75* and *asd-2*, which were up-regulated under DR, and whose target genes were translationally promoted, both have reported activities in RNA binding and splicing (Kuroyanagi et al, 2007, 2013; Ohno et al, 2012; Boateng et al, 2017). We used RNAi for *unc-75* and *asd-2* to determine whether their expression was necessary for extended life span under DR. If their expression is necessary for DR-induced longevity, then we would expect RNAi treatment under to DR to reduce longevity. Knockdown of *unc-75* under DR increased median life span by 11%, suggesting that *unc-75* is not required for DR-induced longevity. However, *unc-75* knockdown resulted in a 47% increase in longevity under AL. RNAi treatment of *asd-2* did not significantly reduce longevity under DR compared with controls. Thus, neither *unc-75* nor *asd-2* were necessary for longevity under DR. However, to our knowledge, unc-75 RNAi was not previously annotated as having a longevity effect under AL.

To investigate potential miRNAs that may influence selective translation, we searched for enrichment of miRNA binding sites among translationally promoted and suppressed genes under DR. Predictions of conserved miRNA family targets within 3′ UTRs were taken from targetscanworm.org (Jan et al, 2011). Targets of miRNA families miR-2/43/250/797, miR-253, miR-58, miR-80/81/82/1835/2209a, and miR-86/785 were significantly enriched ($P < 0.05$, Fisher's exact test) among promoted DPAR genes (Fig 6A). No significant enrichment of targets was found among repressed DPAR genes. The most commonly observed function of miRNAs is silencing gene expression either through degradation of the target or through translational suppression (Catalanotto et al, 2016). Therefore, if these miRNAs are playing a role in promoting translation under DR, we predict that it is because their expression is down-regulated. To examine this, we quantified the expression of each miRNA in the enriched families under DR using qPCR (Fig 6B and Table S9). In response to DR, *mir-58* and *mir-80/81/82* were significantly down-regulated as compared with AL. The other miRNA did not exhibit a significant change in expression. In line with our results, *mir-58* and

**Table 1. Characterization of 3′ UTRs of genes with promoted or repressed polysomal association under DR.**

| | 3′ UTR | | |
|---|---|---|---|
| | **Promoted** | **Repressed** | **All** |
| Mean BP | **148.2** | 170.6 | 180.6 |
| SD BP | 124.2 | 137.9 | 142.1 |
| *P*-value BP | $8.71 \times 10^{-09}$ | 0.406 | n/a |
| Mean GC | 0.267 | 0.279 | 0.271 |
| SD GC | 0.073 | 0.071 | 0.067 |
| *P*-value GC | 0.121 | 0.205 | n/a |
| Mean MFE | **−19.1** | −25.1 | −21.5 |
| SD MFE | 27.0 | 32.9 | 25.6 |
| *P*-value MFE | $7.19 \times 10^{-05}$ | 0.912 | n/a |
| n | 446 | 81 | 3675 |

Promoted, genes with positive DPAR; Repressed, genes with a negative DPAR; All, all quantifiable genes in the dataset; mean GC, the mean GC content of the UTR sequence; avg BP, the mean sequence base pair length; mean MFE, the mean minimum free energy in kcal/mol; n, the number of sequences available for use in the analysis.
Values in bold are significantly (Wilcoxon rank sum test, *P*-value < 0.05) different compared with all quantifiable genes.

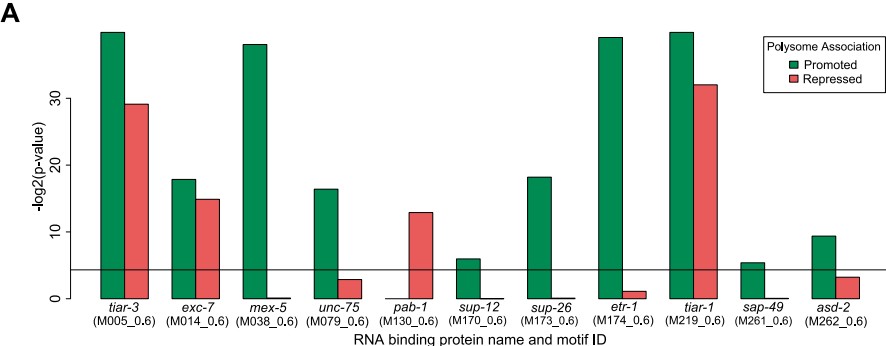

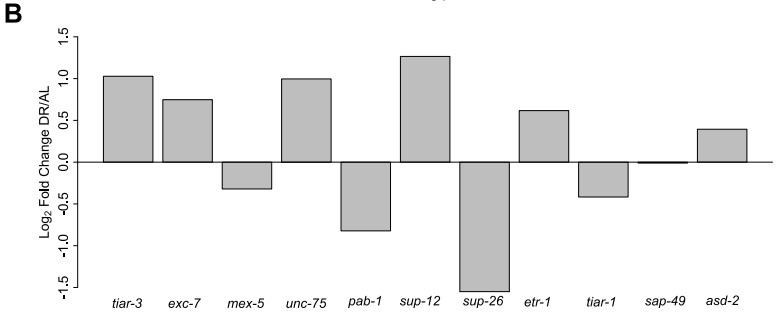

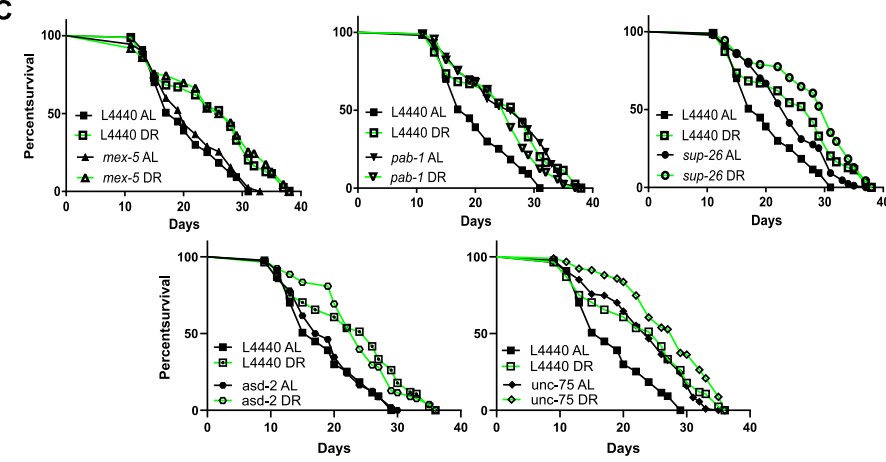

**Figure 5.** ***Trans*-acting factors show enrichment of putative targets among differentially translated genes under DR.**
**(A)** RBP motifs significantly enriched in 3′ UTRs among differentially translated genes. Black line represents significance threshold of $P < 0.05$. The CISBP-RNA database motif ID associated with each RBP is given in parenthesis. **(B)** Regulation of RBP genes with enriched motifs under DR. Fold change in the translated fraction is given. **(C)** RNAi of RBP genes *sup-26* ($P = 1.39 \times 10^{-04}$), *pab-1* ($P = 2.60 \times 10^{-06}$), and *unc-75* ($P = 2.6 \times 10^{-06}$) but not mex-5 ($P = 0.19$) nor asd-2 ($P = 0.585$) extend median life span under well-fed conditions compared with controls (n > 80, log-rank test). Representative life spans are shown; full dataset found in Table S8.

*mir-80* were shown previously to be down-regulated under DR in *C. elegans* (Vora et al, 2013; Kogure et al, 2017). Deletion of *mir-80* recapitulates many of the physiological and longevity benefits of DR (Vora et al, 2013). Furthermore, *mir-58* was recently shown to be partially required for life span extension in a genetic model of DR in *C. elegans* (Zhang et al, 2018). Based on our in silico analysis and the down-regulation of *mir-58* and *mir-80* under DR, they seem to play a role in longevity by regulating the translation of select transcripts.

**Intron retention increases under DR by bacterial dilution**

The alternative splicing (AS) of mRNA can produce gene isoforms that are differentially translated. AS can lead to transcripts with premature termination codons (PTCs) that are targeted for degradation by nonsense-mediated decay (NMD), an mRNA quality control surveillance mechanism that degrades aberrant mRNA.

However, the full range of NMD targets extends beyond aberrant gene products, and some studies suggest that AS and NMD are coupled (AS-NMD) to enable regulation of specific genes via translation (Barberan-Soler et al, 2009; Ge & Porse, 2014). Exactly how NMD targets non-aberrant mRNA is unknown, but these and other studies point to the existence of an underappreciated form of gene regulation.

To assess AS-NMD in the current study, we focused on a type of AS in which an intron is retained in the mature mRNA, which usually leads to out-of-frame translation and one or more PTCs. An average of 1.5% of reads aligned to introns under AL, which was increased to 2.2% under DR ($P = 0.002$, two-way *t* test, Fig 7A). The change in intron retention under DR suggests either diminished NMD, increased AS, or both. Next, we quantified intron retention in the polysome-associated fraction. NMD takes place during a pioneering round of translation, so we expected mRNAs with retained introns to be depleted among polysomes. Indeed, intron retention was lower in

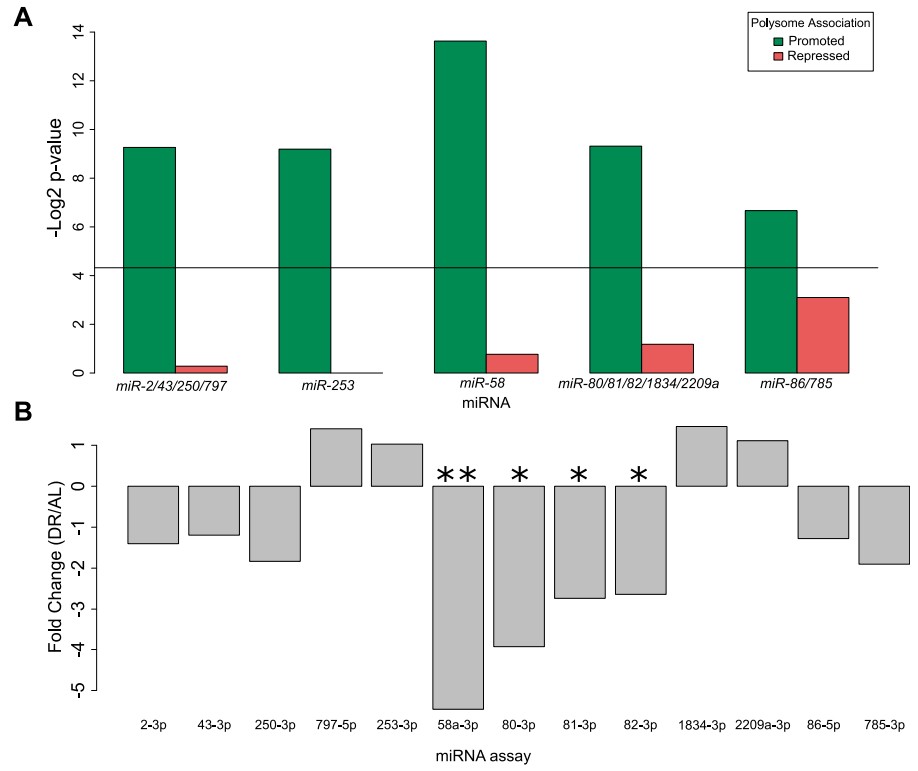

**Figure 6. Enrichment of miRNAs among DPAR genes and their relative expression under dietary restriction.**

**(A)** miRNA binding sites enriched among translationally promoted or repressed genes under DR. Black line represents significance threshold of $P <$ 0.05. Names of the conserved miRNA families are given. **(B)** Expression of the miRNA with motifs enriched in part (A) were quantified under DR. The qPCR analysis indicated that mir-58 and mir-80/81/82 were significantly down-regulated ($P =$ 0.009, 0.022, 0.041, 0.047, respectively; $t$ test. *$P <$ 0.05, **$P <$ 0.01 ) under DR compared AL. The average linear fold change of three biological replicates is plotted.

the translated fraction than the total fraction under both AL (1.0%, $P$-value = 0.002) and DR (1.2%, $P$-value = 0.003). However, the presence of intronic reads in the polysomal fraction suggests that their associated transcripts escaped NMD. Furthermore, although more intron retention was detected in the translated fraction under DR than AL (two-way $t$ test, $P$ = 0.017), the ratio was smaller than that observed between total and translated under AL ($P$ = 0.006, two-way ANOVA). Thus, although there are more retained introns among both total and translated mRNA under DR, there is a decrease in the frequency with which intron-bearing mRNA makes it into the translated fraction. These results suggest that intron retention, like 3′ UTR editing, is a method to inhibit protein synthesis of specific genes during adaptation to DR conditions.

Having established that there are dynamic changes in the global frequency of intron retention under DR, we sought to determine changes in the distribution of intron retention on a gene-by-gene basis. Using a fold-change threshold of 2 and a FDR cutoff of 0.05, 1,029 genes showed increased intron retention and 36 genes exhibited decreased intron retention under DR (Fig 7B). There was a significant overlap in genes with intron retention in the current study with those determined to have intron retention in an *eat-2* genetic model of DR in *C. elegans*, where increased retention of introns was also observed (Fig 7C; Fisher's exact test, $P$ = 0.01; Tabrez et al, 2017). In the translated fraction, 539 genes showed increased intron retention under DR, whereas 42 genes had decreased intron retention (Fig 7B). This suggests that changes in intron retention were because of an increase in the number of genes with retained introns. This also suggests that the increased frequency of introns under DR shown in Fig 7A was not simply because of increased

frequency of introns within a set of genes common to both AL and DR conditions.

Comparison of genes with increased intron retention under DR in the total and translated fractions shows that most, but not all, of those in the translated pool are a minority subset of genes in the total fraction (Fig 7D). Interestingly, 97 genes exhibited a higher level of intron retention among translated mRNA than among total mRNA, suggesting that the retained introns promoted association with translation machinery in these instances (Fig 7D). Results indicated that intron retention was not always associated with exclusion from translation. This suggested that there was more than just nonsense associated with these transcripts, and we further investigated the connection between intron-bearing mRNAs and translation.

To identify dynamic changes in translation of intron-bearing genes between AL and DR, we used DPAR-type analysis similar to that carried out in Fig 2D. Seventy-four intron-retaining genes were associated more with polysomes under DR than AL and 139 were associated less (Fig 7E and Table S10). Those intron-bearing transcripts translationally promoted under DR were enriched for biological processes related to cell localization and signal trans-duction, whereas those translated less under DR were enriched for biological processes including cell cycle, stress response, and telomere maintenance (Fig 7F and G and Table S11). Enrichment of genes governing specific biological process suggests that NMD may be selective in a manner that regulates differential gene expression as opposed to just being a quality control process.

Although we expect the majority of gene products with retained introns to be nonfunctional because of frame shift and/or the

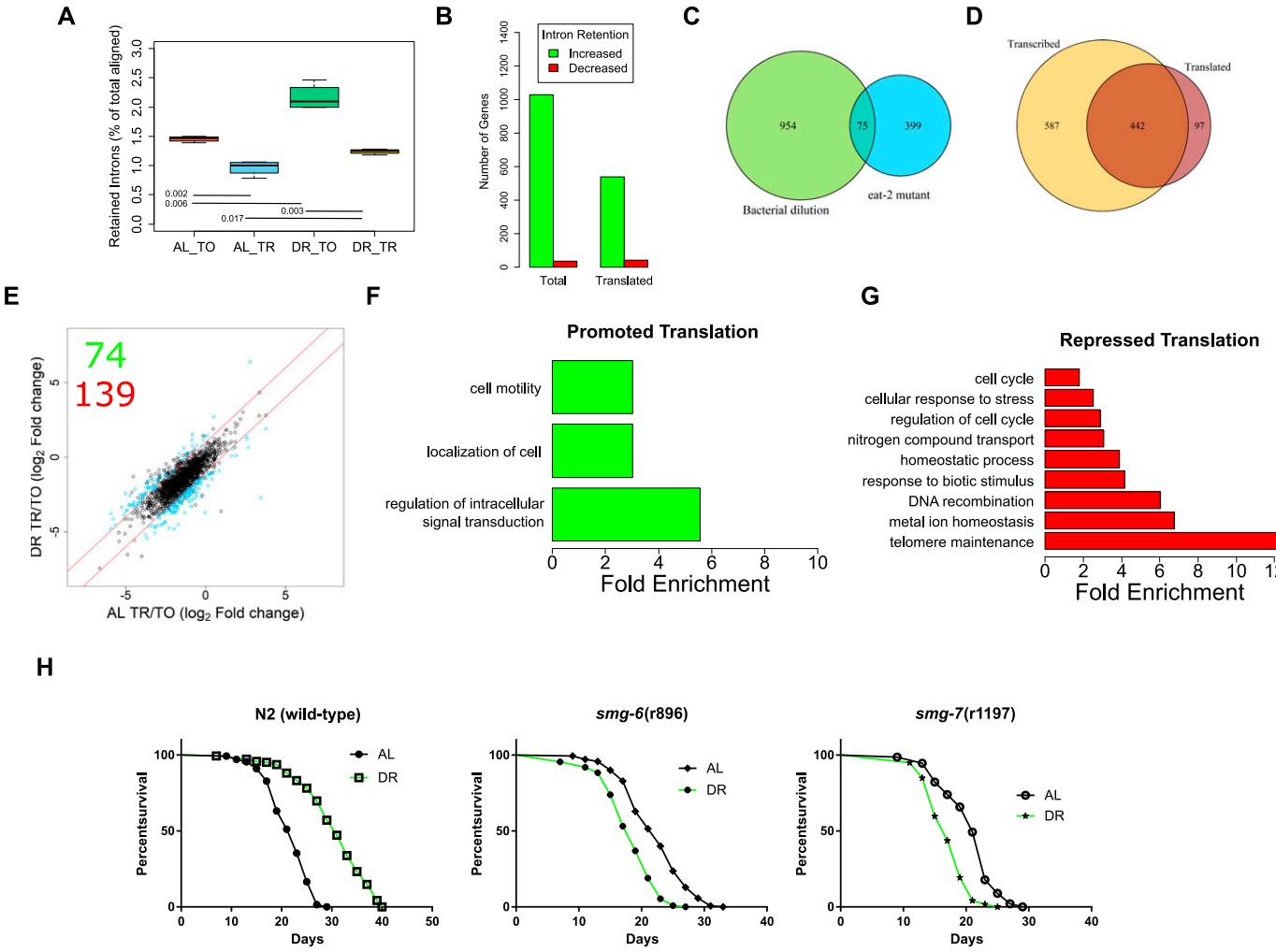

**Figure 7. Intron retention is increased under DR to drive changes in translation.**
**(A)** The reads aligning to introns as a percent of total aligned. *P*-values from two-tailed *t* test, n = 4. **(B)** Number of genes with increased or decreased intron retention under DR in either the total or the translated fraction. **(C)** Venn diagram showing the overlap of genes with intron retention under DR by bacterial dilution in the current study and those identified in the *eat-2* genetic model of DR. **(D)** Overlap of genes with increased intron retention under DR in either the total or the translated fraction. Only genes with >twofold change with FDR >0.05 were considered. **(E)** Retained intron transcripts translationally promoted or repressed significantly (>twofold, FDR < 0.05) under DR. Black circles represent genes with similar translation between diets, blue circles represent genes with twofold or greater DPAR. Red lines delineate the fold twofold change cutoff. Inset, 74 genes were translationally promoted and 139 were translationally repressed. **(F, G)** Biological GO terms enriched among retained intron genes with (F) promoted or (G) repressed translation under DR. *P* < 0.05, modified Fisher's exact test. Functionally similar GO terms from identical groups of genes were manually removed. **(H)** Mutants defective in NMD genes *smg-6* or *smg-7* showed a reduction in life span under DR, in contrast to wild-type worms (*P* < 0.001, long-rank test). Representative life spans are shown; full dataset found in Table S10. TO, total fraction; TR, translated fraction.

presence of PTCs, we sought to examine these assumptions empirically and inspect the structure of genes with retained introns. We examined some of the up-regulated genes more closely to confirm that intron retention was associated with inclusion of one or more PTCs. Table S10 shows the frequency of intron retention for individual genes. We first investigated the basic helix-loop-helix protein (bHLH) *mdl-1* because of its role in regulating *C. elegans* life span (Nakamura et al, 2016) and because it was previously reported to undergo intron retention (Tabrez et al, 2017). The predicted CDS of the intron-bearing isoform of mdl-1 contained an out-of-frame PTC occurring after its annotated bHLH domain (Fig S4A). Therefore, although the retained intron isoform would encode for a truncated protein, it could still have a function through the expressed bHLH domain.

The gene F13G3.6 was investigated because it was reported to have a PTC-bearing isoform whose expression is dependent on the activity of the NMD gene *smg-2* in *C. elegans* (Son et al, 2017). This gene is similar to human B3GNTL1, which plays a role in elongating N-glycans on mucin proteins. Exons upstream of the intron contained a glycosyl transferase family two domain. Interestingly, the retained intron results in a single PTC with no frame shift (Fig S4B). In such instances, it is possible that translational read-through of the PTC occurs to produce a full-length product (Oren et al, 2014).

Finally, we examined the gene encoding selenoprotein *trxr-1* as it mediates healthy longevity in response to treatment with the organic form of selenium (Chang et al, 2017) and only retained a portion of the annotated intron. The retention event resulted in

addition of 6 base pairs (bp) to the 5′ end of exon 5, which did not induce a PTC or frame shift (Fig S4C). Therefore, while unlikely to be directly regulated by NMD, this isoform may represent a new, previously annotated isoform of *trxr-1* that is better translated under DR.

### NMD genes *smg-6* and *smg-7* are required for DR-mediated life span extension

In *C. elegans*, NMD is governed by the activity of several genes, including *smg-1/2/3/4/5/6/7*, which are responsible for the detection of aberrant transcripts, releasing the stalled ribosome, and degrading the transcript (Hug et al, 2016). The expression of several smg genes were altered in the DR mRNA-seq dataset (Fig S4D). Given that the translation of NMD-prone transcripts seemed to be regulated by DR and that NMD has been shown to play a role in ILS-mediated longevity (Son et al, 2017), we hypothesized that NMD plays a role in DR-induced longevity. We tested the importance of NMD in longevity using mutants defective in *smg-1/2/3/4/5/6/7* subject to AL or DR starting at adulthood. Interestingly, the lines *smg-6(r896)* and *smg-7(r1197)* were shorter lived under DR than AL (*P*-value < 0.0001 for both; log-rank test), demonstrating their requirement for DR-induced longevity (Fig 7H). In comparison, the other smg mutants (*smg-1/2/3/4/5*) were longer lived under DR than AL, albeit this extension was attenuated compared with wild-type worms (Table S12). The requirement of *smg-2* for the full effects of DR-induced longevity in the *eat-2* mutant and by bacterial dilution was previously shown (Tabrez et al, 2017). The genes *smg-6* and *smg-7* are each thought to participate in NMD by degrading the transcript after it has been detected and the ribosome released. Although *smg-6* is thought to degrade transcripts directly by its endonuclease activity, *smg-7* is thought to degrade transcripts by recruiting deadenylation factors. Together, these observations raise the possibility that the selective degradation of NMD-prone transcripts by *smg-6* and *smg-7* can dictate which transcripts are translated under DR to influence longevity.

## Discussion

Translation is expensive with respect to the energy and raw materials that it consumes. Low nutrient levels lead to low global translation. However, our understanding of how the stoichiometry of protein synthesis for specific individual gene products changes under DR through selective translation is limited. To elucidate this process, we sought to identify translationally regulated genes and potential mechanisms of posttranscriptional regulation that influence their translation. Using a combination of polysome profiling and mRNA sequencing, we generated diagnostics of the transcriptome and translateome. In silico analysis was used to assess the relationship between these diagnostics and how this relationship is changed when nutrients are restricted. It also provided a means by which to probe differentially expressed genes for annotated *cis*-elements and corresponding *trans*-acting factors, generating a roadmap for future analyses testing mechanisms of posttranscriptional regulation that contribute to differential expression. The use of RNAi allowed us to determine which

differentially expressed genes and cellular processes, like NMD, are important contributors to the increased life span associated with DR.

An important early step in trying to understand how somatic protection is conferred by a health-promoting intervention like DR is to understand how expressions of genes change to meet the requirements of adaptation. Three genes among the 100 showing the greatest transcript-level decrease in expression under DR increased life span when knocked down with RNAi under AL. *fbxb-24* and *sdz-33* encode highly related F-box–containing proteins. The F-box motif was first observed as a constituent of Skp, Cullin, F-box (SCF) ubiquitin–ligase complexes and helped confer specificity to proteolytic substrates (Skowyra et al, 1997). Since then, many other genes encoding F-box motifs have been identified. The motif is structured with about 50 amino acids that function in protein–protein interactions, which are highly abundant in *C. elegans* (Kipreos & Pagano, 2000). The third gene was D1086.3, a gene of unknown function that was also the only one tested for life span that was also selectively down-regulated via translation.

Selective translation of stress responsive genes important for longevity has been previously described in *C. elegans* ILS mutants (McColl et al, 2010) and in a model of restricted nutrient-responsive translation (Rogers et al, 2011). In *Drosophila melanogaster*, selective translation enhanced synthesis of genes encoding mitochondrial components on a form of DR involving restriction of yeast extract (a primary source of protein; Zid et al, 2009). Inhibition of the nutrient-responsive TOR complex in mouse embryonic fibroblasts selectively suppressed ribosomal genes and promoted translation of several transcription factors, including *Foxo1* (Thoreen et al, 2012). We observed translational suppression of ribosomal genes in the current study and other genes involved in translation (Fig 2G and Table S3). Two of these genes, *iff-1* and *pab-1*, are translation initiation factors that increased life span when knocked down in fully fed wild-type animals.

In addition to direct regulators of translation, other repressed DPAR genes involved in life span regulation encode chaperones. The largest increase in median life span was observed for *daf-21*, the *C. elegans* ortholog of human HSP90. Although it has been previously reported to increase life span (Horikawa et al, 2015; Sutphin et al, 2017), this is the first study to demonstrate its translational regulation in the context of DR. DAF-21/HSP90 serves to promote growth and reproduction by keeping growth-related kinases and transcription factors in a near-native conformation that facilitates their activation (Mitra et al, 2016). HSP90 also sequesters the HSF-1 heat shock transcription factor, inhibiting its ability to trimerize and up-regulate expression of heat shock protein genes (Zou et al, 1998). The translational down-regulation of *daf-21* under DR makes sense given that there should be less need for activation of growth-related kinases and transcription factors in low-energy conditions when nutrients are depleted. The chaperonin gene, *cct-4*, was translationally down-regulated under DR and increased life span upon knockdown despite the fact that loss of *cct-4* is associated with a reduction in *skn-1* transcriptional activity (Kahn et al, 2008). *skn-1* is the only worm ortholog of mammalian nuclear factor-erythroid-related factor (Nrf) transcription factors and functions in the p38 MAPK pathway to regulate the oxidative stress response. Its transcriptional activity is inhibited by ILS (Tullet

et al, 2008), and it is considered to be a longevity factor by promoting somatic maintenance. It is unclear why low *cct-4* translation supports increased life span given its positive correlation with at least one *skn-1* target, *gst-4*. It is possible that other *skn-1* targets are not similarly affected by *cct-4* expression.

Translationally down-regulated genes D1086.3 and *kbp-1* were, for the first time, identified as life span regulators in our RNAi screen. Interestingly, we did not observe life span extension in our RNAi screen for certain repressed DPAR genes previously shown to regulate life span. For example, expression of *skr-1* and *ncbp-2* are positively associated with longevity, whereas expression of *cyc-2.1* is negatively associated with longevity but had no effect when knocked down using RNAi in the present study. The importance of *skr-1* for increased life span is specific to attenuated ILS in *daf-2* mutant backgrounds of multiple alleles tested and does not regulate life span in an *eat-2(ad1116)* genetic model of DR (Ghazi et al, 2007). RNAi targeting the nuclear cap-binding protein gene *ncbp-2* was only shown to shorten life span when initiated during development, which can also result in severe defects and lethality (Büssing et al, 2010). *cyc-2.1* encodes a component of the electron transport chain, which only extends life span in N2 wild-type animals when it is knocked down during development. Thus, it is not surprising that these genes did not regulate life span in our longevity screen that began during adulthood.

### A novel association between RNA editing and translation under DR

Although RNA editing frequencies were previously shown to be regulated by neuronal stimulation (Tariq et al, 2013), regeneration (Witman et al, 2013), and hypoxia (Ben-Zvi et al, 2013), this is the first study to reveal widespread changes in editing because of nutrient limitation. A-to-I RNA editing increased under DR, especially within the 3′ UTRs, and such editing was associated with translational suppression. Conversely, although editing events occurred in the CDS, they were less frequent and were not associated with translational suppression. Together, these results indicate that RNA editing events are dynamic and likely to play a role in the adaptive response to DR.

Although a link between editing of individual genes and longevity remains to be established, polymorphisms in mRNA editing genes have been associated with longevity in humans (Sebastiani et al, 2009). The same study also demonstrated that loss the orthologous editing genes in *C. elegans* were required for longevity (Sebastiani et al, 2009). This raises the possibility that mRNA editing regulates the selective translation of transcripts with inputs to longevity. A provocative example of this in the current dataset was the editing of the 3′ UTR of *eif-2alpha*. Furthermore, the enrichment of RNA editing sites within 3′ UTRs was shown to overlap frequently with miRNA binding sites (Gu et al, 2012). Thus, RNA editing may influence expression by introducing or removing miRNA binding sites.

### RBPs and translation regulation under DR

Our screening of RBP motifs implicated *pab-1* in the regulation of translationally promoted genes under DR. This gene encodes for the cytoplasmic poly-A–binding protein, referred to as PABPC1 in mammals. It has several molecular functions associated with it, the most prominent of which is its ability to bind to poly-adenosine stretches in the 3′ UTR of mRNA and then to associate with factors in the 5′ UTR to promote translation initiation. PABPC1 also regulates the expression of specific mRNA by binding to regions outside of the polyA tail (Smith et al, 2014). This mRNA-specific regulation can be mediated by interactions with miRNA (Flamand et al, 2016), participation in NMD (Brook & Gray, 2012), or through association with the 3′ UTR instead of the polyA tail (Burgess & Gray, 2010). Thus, *pab-1* may be important for multiple forms of differential translation under DR.

### Translational preference for *trans*-spliced products is reversed for the subset of SL genes in operons during adaptation to DR

5′ UTRs of *C. elegans* transcripts are frequently *trans*-spliced, which is a feature not only common to nematodes but also to flatworms, protozoa, sponges, cnidarians, chaetognaths, crustaceans, rotifers, and tunicates (Pouchkina-Stantcheva & Tunnacliffe, 2005; Zayas et al, 2005; Hastings, 2005; Zhang et al, 2007; Marlétaz et al, 2008). SL sequences in nematodes bear trimethylguanosine caps ($m_{2,2,7}$G; Liou & Blumenthal, 1990), in contrast to native 5′ UTRs, which contain a monomethylguanosine ($m_7$G) cap (Sonenberg, 2000). Under well-fed conditions, we found that *trans*-spliced mRNA was translated better than mRNA with the native 5′ UTR, supporting previous analysis (Yang et al, 2017). Although this pattern was maintained under DR, it was modestly but significantly attenuated for SL1 spliced genes (Fig 3B).

Interestingly, despite the preference for SL-containing mRNA to be translated under normal conditions, both SL1 and SL2 genes were underrepresented among promoted DPAR genes (Fig 3C and D). This was also true for polycistronic genes contained in operons, which represent a subpopulation of *trans*-spliced products that accounts for ~17% of all protein-coding genes in *C. elegans* (Blumenthal & Gleason, 2003). Unlike bacterial operons, nematode polycistrons are separated posttranscriptionally and translated separately. The first gene within an operon is spliced to SL1, and subsequent genes are spliced to SL2. Interestingly, nearly half of the DPAR genes down-regulated in the transition from AL to DR conditions reside in operons. Although it is not clear how they are down-regulated, co-regulation of multiple genes by a single promoter could represent an optimization of gene expression for growth and reproduction under favorable conditions. Unfavorable conditions may then necessitate differential expression of genes under the same promoter, which could be achieved through translational regulation. However, because of the differences in the broader trend for translational regulation of trans-spliced genes from the subset of those in operons, we surmise that the presence of SLs, alone, cannot account for the differential translation of operonic genes under DR.

### DR increases intron retention products and differentially regulates NMD targets

We found evidence that DR increases the prevalence of intron retention and alters NMD. Furthermore, we found that certain NMD

genes that play a role in target specificity and degradation are required for the benefits of DR. How AS-NMD is integrated to influence differential gene expression remains to be fully elucidated. With respect to splicing, there is a limited but growing body of evidence pointing to changes in AS according to dietary and metabolic conditions. For example, DR is associated with changes in AS for individual metabolic genes in mammals (Gallardo et al, 2005; Fernández et al, 2009; Kaminska et al, 2014). Furthermore, hepatic transcripts in caloric restricted primates exhibited increased alternative exon use, including genes related to fatty acid metabolism (Rhoads et al, 2018). Conversely, obesity is associated with diminished expression of several splicing factors in liver and skeletal muscle in a manner that contributes to increased lipogenesis (Pihlajamäki et al, 2011). In a recent non-DR–related study of splicing in the tea plant *Camellia sinensis*, the predominant form of AS was intron retention, which occurred in genes related to the spliceosome and mRNA processing (Zhu et al, 2018). That regulators of splicing are, themselves, highly subject to alternative processing is conserved in animal studies. In mammals, AS is regulated by members of the SR protein family that alternatively splice a portion of their own transcripts (Lareau et al, 2007). These splice variants are normally targeted by NMD to limit total SR protein expression. Our analysis of intron retention suggested that coupling of AS-NMD is modulated under DR, in part, by changing the selectivity toward which transcripts are detected and degraded. Observations are in line with those obtained from cell culture exposed to rapamycin, a drug that inhibits the DR-related TOR pathway, in which the selectivity of NMD toward transcripts was also modulated (Martinez-Nunez et al, 2016). Taken together, studies point to an adaptive mechanism of DR that functions by controlling productive splicing and turnover.

The idea of regulated AS-NMD was previously invoked to explain differential expression during development in *C. elegans* (Barberan-Soler et al, 2009). Recently, Tabrez et al (2017) used a genetic model of DR in *C. elegans* and identified more than 400 differential intron retention events. The authors suggested that AS under DR is needed to diversify the proteome in a way that is conducive for longevity and additionally requires NMD to fine-tune gene expression. The same *eat-2* mutant was found to have increased splicing in the early, healthy part of life (Heintz et al, 2016). Thus, although it is clear that AS and intron retention is modulated by DR in a way that may affect longevity, the roles of AS-NMD on differential gene expression are, as yet, unclear.

This study was designed to characterize the posttranscriptional landscape of gene expression in response to DR and then survey the mechanisms that may govern this regulation. Our data provide evidence that DR invokes selective translation by acting through miRNA, RBPs, 3′ UTR length, RNA editing, and intron retention (Fig S5). One limitation to this approach was the use of whole worms as opposed to specific cell lines or tissues. Although the use of whole worms allows for the quantification systemic changes on translational regulation or effects occurring in large tissues like the intestine, tissue-specific effects, like RNA editing in the neurons, are likely lost by averaging the effect over all other tissues. The use of tagged RBPs, like poly-A–binding protein, expressed by tissue-specific promoters has been used in *C. elegans* to quantify tissue-specific gene expression in total mRNA pools (Blazie et al, 2015, 2017). Future studies may rely on similar technology to identify genome-wide tissue-specific patterns of differential processing and expression.

## Materials and Methods

### Nematode culture and polysome profiling

*Caenorhabditis elegans* strains were cultured at 20°C and maintained on normal growth medium (NGM) plates seeded with OP50. Developmentally synchronous animals from a 4-h egg lay were transferred to AL ($10^{11}$ CFU/ml) or DR ($10^{11}$ CFU/ml) reaching day 1 of adulthood. To inhibit bacterial growth, AL and DR plates contained 25 mg/ml carbenicillin. Because the DR conditions used in this study inhibit reproduction, 50 µg/ml FUdR was used to inhibit egg production equally under DR and AL. Polysome profiling was performed as previously described (Pan et al, 2007) after 4 d under these culture conditions. 100 µl of pelleted worms were homogenized on ice in 350 µl of solubilization buffer (300 mM NaCl, 50 mM Tris–HCl [pH 8.0], 10 mM $MgCl_2$, 1 mM EGTA, 200 g heparin/ml, 400 U RNAsin/ml, 1.0 mM phenylmethylsulfonyl fluoride, 0.2 mg cycloheximide/ml, 1% Triton X-100, and 0.1% sodium deoxycholate). After homogenation, an additional 700 µl of solubilization buffer was added and the sample was incubated on ice for 30 min. Debris was pelleted by centrifugation at 20,000$g$ for 15 min at 4°C. Of the resulting supernatant, 900 µl was applied to the top of a 10–50% sucrose gradient in high salt resolving buffer (140 mM NaCl, 25 mM Tris–HCl [pH 8.0], and 10 mM $MgCl_2$) and centrifuged in a Beckman SW41Ti rotor at 38,000 rpm for 90 min at 4°C. Gradients were fractionated using a Teledyne density gradient fractionator with continuous monitoring of absorbance at 252 nm. RNA was extracted from ploysome fractions and unfractionated samples using Trizol LS reagent according to the manufacturer's protocol (Invitrogen Corp.). Eukaryote total RNA Nano chips were used in an Agilent Bioanalyzer for quality control of the resulting RNA.

### Alignment of reads to the *C. Elegans* genome

Alignment of RNA-seq reads from each biological replicate to *C. elegans* genome version WS220 was performed using the splice junction mapper TopHat (ver 2.0.8.b) (Kim et al, 2013). TopHat was supplied with transcript annotations from ENSEMBL version 66 (Flicek et al, 2011), so that preference of aligning reads is given to the transcriptome first (remaining reads are converted to genomic mappings and merged with the final output). Default settings of Tophat were used with the following exceptions: the mean mate inner distance was set to 200 bp, microexon search was enabled, and the library type was set to "fr-unstranded" to indicate that unstranded paired end reads were used.

### Differential expression

Aligned reads were counted per gene using the python script HTseq (Anders et al, 2015). Differential expression and dataset normalization was performed using the Bioconductor package edgeR

(Robinson et al, 2010). Normalization in edgeR adjusted for RNA composition to ensure that highly expressed genes which consume a large portion of the RNA pool did not result in the undersampling of other genes. In addition, as our data set contained multiple factors, dispersion of the gene counts were estimated tagwise using the Cox–Reid profile-adjusted likelihood method (Cox & Reid, 1987). Only genes with average counts per million (CPM) of eight or greater across all conditions were considered reliably quantifiable for differential expression. A threshold of eight CPM was chosen as it was the point where the trend in the biological coefficient of variation approached the common variation. Differential expression was calculated pairwise between groups using a general linear model, and the resulting $P$-values were adjusted for multiple testing using the Benjamini–Hochberg method (Benjamini & Hochberg, 1995). The DPAR was calculated using the following GLM using edgeR's makecontrast() command as follows: (DR_TR - DR_TO)-(AL_TR - AL_TO).

### miRNA relative quantification

Samples were day 4 adult N2 worms subjected to AL food or DR by bacterial dilution starting at adulthood. Biological triplicates were performed for each treatment. Worms were washed off from NGM plates with 1 ml of M9 buffer and allowed to gravity settled. Settled worms were washed twice with M9 to remove bacterial contamination and flash frozen before storage at –80°C. RNA extraction with Trizol Reagent (Life Technologies) was performed according to the manufacturer's instructions until phase separation. The aqueous phase was then used as starting sample for further purification using the *RNA Clean & Concentrator-5* system (Cat. No. 1016; Zymo Research) following the manufacturer's protocol recommended for *Trizol Clean-up*. RNA quality was analyzed using the NanoDrop ND-1000 Spectrophotometer (software v3.8.1). miRNA-specific cDNA was prepared using the *qScript microRNA cDNA Synthesis Kit* (Cat. No. 95107-025; Quantabio) following the manufacturer's protocol using 316 ng. The *C. elegans* mature miRNA sequences were obtained from miRbase (http://www.mirbase.org/index.shtml). All the miRNA members of families predicted to be enriched among DPAR genes (Fig 6B) were included as were their 5 prime (5p) or 3 prime (3p) product partners. In addition, primers for both the 3p and 5p products of *mir-2*, *mir-46*, and *mir-47* were included as potential normalizers as they were previously accessed in *C. elegans* to be stably expressed under stress (Kagias et al, 2014). qPCR reactions consisted of 1× rAmp qPCR Green Master Mix (Denville Scientific/Thomas Scientific, 5 $\mu$m forward primer, 5 $\mu$m PerfeC$_T$a Universal PCR Primer, and 3.16 ng RNA equivalents of cDNA all in a 10-$\mu$l reaction volume. qPCR data were collected using the LightCycler 480 (SW 1.5.1; Roche) equipped with a 384-well block. Thermal cycling was as follows: one step hold at 95°C for 2 min, 40 cycles of 95°C for 5 s/60°C for 20 s (data collection at 60°C), and one step melt with continuous data collection for melting curve analysis. Cycle threshold values from qPCR were analyzed with a modified version of Global Pattern Recognition (GPR) (Akilesh, 2003), and threshold cutoff setting was set to 35. GPR generates aggregate $P$-values corresponding to the likelihood that the gene is changing between treatments based on all using other assays as normalizers.

The reported fold change was calculated in GPR by using 10 normalizer assays. These normalizers were the 10 assays with the lowest average variation across all samples. The results of all considered miRNA assays are supplied in Table S9.

### Expression profile clustering

Genes with a significant twofold or greater difference in relative expression between total and polysome-associated mRNA under DR but not AL were clustered together. Before clustering, the expression of each gene among the four experimental groups in was centered around its mean expression in the dataset. Clustering was performed with the Partitioning around Medoids algorithm using the pam() function and evaluated using silhouettes widths (Rousseeuw, 1987) with the silhouette() function, both contained in the R package "Cluster". The maximal number of clusters to use was determined by reiteratively running the Partitioning around Medoids algorithm while sequentially increasing the number of clusters. Based on the experiences of Kaufman & Rousseeuw (2005), a silhouette width cutoff of 0.25 was used as the lower limit for a cluster to have substantial structure. The highest cluster number among all iterations with all clusters having substantial structure was three.

### RBP motif enrichment

Experimentally determined and homology inferred position weighted matrices for RNA-binding motifs of *C. elegans* proteins were downloaded from the CISBP-RNA database (Ray et al, 2013). Enrichment for each motif among genes translationally promoted or suppressed under DR was calculated using the R package "PWMEnrich". Background correction for nucleotide composition was applied with the function makePWMLognBackground() using 3′ UTRs from all quantifiable genes. 3′ UTRs were exported from the *C. elegans* genome version WS220; 3′ UTRs shorter than 12 bp were excluded from the analysis.

### miRNA motif enrichment

Predicted targets of conserved miRNA families among nematodes were imported from http://www.targetscan.org/ release 6.2 (Jan et al, 2011). Only targets with a probability of conserved targeting ($P_{ct}$) >0.8 were considered. Enrichment of miRNA binding sites among translationally promoted or suppressed genes compared with all quantifiable coding genes was calculated using Fisher's exact test.

### SLs

Annotations of the predominant SL present in *C. elegans* transcripts were taken from (Allen et al, 2011) Table S2. Unannotated genes were considered not to undergo *trans*-splicing and have their native 5′ UTRs. Comparison of translational regulation between AL and DR of genes with native, SL1, or SL2 5′ UTRs was performed using the Wilcoxon rank sum test with continuity correction.

## Intron retention

Reads aligning to intronic or exonic sequences were counted for each biological replicate using HTseq (Anders et al, 2015) based on transcript annotations from ENSEMBL version 66. A minimum of four reads aligning to introns within a gene was required to include in the analysis. The amount of global intron retention was expressed as the percent of reads aligning to introns over all aligned reads. Significant differences in global intron retention were determined using *t* test. To calculate intron retention occurring between diets in the total or translated fraction for individual genes, the Wilcoxon rank sum test was used. To detect genes with intron retention differing between RNA fractions in a way dependent on DR, a linear model was used to determine significant interaction effects of diet and RNA fraction on intron retention. This was achieved using the lm() function in R to model the interaction effect of diet and RNA fraction on intron retention for each gene. Multiple testing was corrected for using the Benjamini–Hochberg method.

## GO term enrichment

Enrichment of biological terms among gene lists was performed using DAVID (version 6.7) (Huang et al, 2008, 2009). The category GOTERM_BP_FAT was used for annotations, and the list of quantifiable coding genes in our dataset was used as the background. For visualization, similar enriched GO terms based on the same subset of genes were manually removed to reduce redundancy; supplemental tables contain the full results. Genes lists submitted to DAVID analysis often had low conversion rates (<80%); therefore, it is likely that enriched terms will change as annotations become more complete.

## Statistical analysis

Spearman correlations between gene abundance in the total and translated fractions was calculated using the rcorr() function in the R package "Hmisc". The distribution of fold change in gene abundance between the total and translated fractions was compared between AL and DR conditions using a two-sample Kolmogorov–Smirnov test.

## Life spans and worm strains

All worm cultures were maintained at 20°C. Worms were subjected to RNAi treatments starting at day 1 of adulthood to avoid potential effects on development. To prevent contamination by progeny, FuDR was used at a concentration of 0.5 mg/ml in the plate. Worms were scored every other day for survival. Worms were scored as alive if movement was evident after gentle prodding of tail and head with a platinum worm pick. Worms with vulvar bursts or that had crawled off media were censored. Survival was plotted as Kaplan–Meier survival curves in GraphPad Prism (version 5) and analyzed by log-rank test using the "survival" package (version 2.44-1.1) in R. Three biological replicates were performed for each life span. The following strains were used in this study: N2, TR1331, CB4043, CB4354, CB4355, TR1335, TR1336, and TR2230.

## RNA interference by feeding and DR

RNAi bacteria strains were cultured as previously described (Kamath et al, 2001). Worms were fed RNAi strains starting at day 1 of adulthood. Bacteria expressing dsRNA included: empty vector L4440 (Addgene), those from the Ahringer library or if not available, from the Vidal ORF library (Source BioScience). For life spans involving both RNAi and DR, the worms were first exposed to regular concentrations of RNAi bacteria for 2 d before transferring to peptone-free nematode growth media plates containing OP50 at the appropriate concentrations for AL ($10^{11}$ CFU/ml) or DR ($10^{11}$ CFU/ml). For the polysome profiling, worms were grown to maturity on OP50 until transferring to AL or DR plates as above.

## RNA editing sites

To search for potential adenosine to inosine RNA editing sites, mRNA-seq reads from each biological replicate were compared with the reference genome WS220 for the presence of A to G SNPs using REDItools (Picardi & Pesole, 2013). The last three bases from each were trimmed, and only uniquely mapped reads from properly paired reads were considered. In addition, for each biological replicate, a minimum of five reads supporting the variation and an editing frequency greater than 10% was required. Significant differences in editing frequencies between experimental groups were determined using the Wilcoxon signed rank test.

## GSEA

Comparison of differentially expressed transcripts in the total fraction of our dataset with those detected in a microarray experiment published by Honjoh et al (2009) was performed using GSEA version 2.0.9 (Subramanian et al, 2005). The processed data from the manuscript were downloaded from Array Express under the genome accession number GSE9682. The processed data were reformatted for GSEA, and Affy IDs were replaced with the gene ID for their matching transcript using Ensembl's Biomart function. Of the 22,626 affy IDs, 21,459 had a gene name assigned to them; those Affy IDs without a corresponding gene ID were not included in the analysis. The GSEA was run with 10,000 permutations based on shuffling the gene set, using the "weighted" enrichment statistic and "signal to noise" metric for ranking genes.

## 5′ and 3′ UTR characterization

5′ and 3′ UTRs were characterized separately based on length, GC content, and minimum free folding energy. Three groups of genes were considered: genes translationally promoted, genes translationally suppressed, and all quantifiable genes. Gene sequences from the 5′ or 3′ ends were exported from ENSEMBL version 66 using the getSequence() function in the biomaRt R package. As genes often encode for multiple isoforms, 3′ and 5′ UTR sequences from the predominately expressed isoforms across the experiment was used for analysis. Quantification of isoform expression was performed using cuffdiff with default parameters (version 2.1.1), and the predominantly expressed isoform was defined as the one with the highest average FPKM in the dataset. Fractional GC content and

sequence length were calculated using the alphabetFrequency() function from the biostrings package. The minimum free folding energy was calculated using the RNAfold function of the Vienna RNA package (version 2.4.8) with default settings. The characterized values for translationally promoted and suppressed genes were compared with all quantifiable genes using the Wilcoxon rank sum test.

## Data Availability

The raw and processed high-throughput sequence data from this publication have been deposited to the Gene Expression Omnibus database https://www.ncbi.nlm.nih.gov/geo/ under accession GSE119485.

## Supplementary Information

## Acknowledgements

The authors would like to thank Ben King, PhD, and Joel Graber, PhD, for bioinformatics advice and George Sutphin, PhD, for constructive criticism of the manuscript. This work was supported by grants from the National Institute on Aging of the National Institutes of Health (R21AG056743) and by the Ellison Medical Foundation (AG-NS-1087-13), both to AN Rogers. In addition, this project was supported by Institutional Development Awards (IDeA) from the National Institute of General Medical Sciences of the National Institutes of Health (grant numbers P20GM0103423 and P20GM104318, respectively). Some strains were provided by the CGC, which is funded by the NIH Office of Research Infrastructure Programs (P40 OD010440).

### Author Contributions

JA Rollins: data curation, formal analysis, methodology, writing—original draft, writing—review and editing.
DJ Shaffer: methodology.
SS Snow: methodology.
P Kapahi: conceptualization.
AN Rogers: conceptualization, funding acquisition, writing—review and editing.

### Conflict of Interest Statement

The authors declare that they have no conflict of interest.

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
