## [Reviewer comments · Life Science Alliance]

Life Science Alliance

Dietary restriction induces post-transcriptional regulation of longevity genes.

Jarod Rollins, Daniel Shaffer, Santina Snow, Pankaj Kapahi, and Aric Rogers

DOI: <https://doi.org/10.26508/lsa.201800281>

Corresponding author(s): Jarod Rollins, MDI Biological Laboratory

Review Timeline:

Submission Date:	2018-12-15
Editorial Decision:	2019-02-13
Revision Received:	2019-06-05
Editorial Decision:	2019-06-18
Revision Received:	2019-06-19
Accepted:	2019-06-20

Scientific Editor: Andrea Leibfried

Transaction Report:

February 13, 2019

Re: Life Science Alliance manuscript #LSA-2018-00281-T

Dr Jarod A Rollins
MDI Biological Laboratory
159 Old Bar Harbor Road
Bar Harbor, Maine 04672

Dear Dr. Rollins,

Thank you for submitting your manuscript entitled "Dietary restriction induces post-transcriptional regulation of longevity genes." to Life Science Alliance. The manuscript was assessed by expert reviewers, whose comments are appended to this letter. I sincerely apologize for the delay in getting back to you. We were promised a third review on your work and the reviewer needed more time. Since we still haven't received this third review, we now decided to move ahead. In case the report gets still submitted, I will forward it to you.

As you will see, both reviewers appreciate your work but think that your conclusions need further support. We would like to invite you to submit a revised version of your work, addressing the individual concerns of the reviewers. Importantly, your conclusions need validation with alternative methods. Furthermore, additional lifespan analyses should get included and both reviewers also note a problem with the RNAi assay being coupled to food restriction which needs to get solved.

Thank you for this interesting contribution to Life Science Alliance. We are looking forward to receiving your revised manuscript.

Sincerely,

B. MANUSCRIPT ORGANIZATION AND FORMATTING:

Reviewer #2 (Comments to the Authors (Required)):

Rollins et al. report here an extensive analysis on the transcriptional and translational regulation induced by bacterial food dilution in *C. elegans*. The differentiation between transcriptome and

translatome regulation with analysis in the same samples appears novel to the DR field. The authors correlate gene expression changes with the abundance of mRNAs in the polysome-associated mRNA fractions. As a consequence, they establish a list of mRNAs with translational suppression or promotion by DR. In their further study they highlight a few novel DR longevity genes and investigate functional analysis. To add mechanistic details for the suppression or promotion of translation in DR, they in silico investigate mRNA processing events affecting translatability such as location of genes in operons, trans-splicing, RNA editing, intron retention and nonsense-mediated decay of their previously selected genes with translational regulation for involvement in promoting DR-mediated longevity.

This manuscript adds to the understanding of the mechanisms by which DR might promote healthier aging. Transcriptome analysis of various DR protocols has been reported multiple times, but most published studies were only focused on gene expression changes. This work focuses on the mRNAs actually translated, adds different aspects of post-transcriptional regulation to the mechanism and constitutes a valuable resource to the community about the effects of DR in *C. elegans*.

However, the manuscript is focused on in silico analysis only, mostly descriptive and correlative, with little experimental laboratory validation to support the computational analysis and claims are often of speculative nature. Literature is appropriately discussed; although at times more recent studies should be included in the discussion. For this reviewer, the missing experimental validation and the limited technical details reduce the excitement for this study.

Major comments and questions:

- 1) What is the reasoning behind comparing the data to IF instead/in addition to comparing to other DR transcriptome analyses? How do the gene expression results compare to other DR transcriptome analyses?
- 2) Since DR slows global translation rates, how does this affect polysome profiling at a defined time point and comparison to samples with a different translation rate? Does this impact the DPAR? Different translation rates should be discussed.
- 3) The number of quantified annotated coding genes is rather low. Is there a technical explanation?
- 4) What is the number of replicates for the lifespan results in Table 1? Some of the lifespan increases are less than 10% and 3 replicates should be presented for statistical relevance to support the claim for these genes and their potential role in DR longevity. Further, what is the reasoning for using FUDR in fully fed conditions? The use of FUDR is controversial in the *C. elegans* field as it has been shown to prolong lifespan by itself and the authors should show the lifespan results without the use of FUDR. Did the authors validate gene expression levels of the novel DR longevity regulators by alternative methods? Have the authors assessed the efficiency of the RNAi knockdown? Lifespan increase for several genes in Table 1 is much less than the longevity seen with DR itself. In order to really prove that these are novel longevity regulators, I would suggest to perform the converse experiment and increase the concentration of respective genes in a DR background and test if DR then fails to extend lifespan.
- 5) How does food dilution affect RNAi knockdown in lifespans for RNA binding proteins? Do you achieve sufficient knockdown with reduced food intake? Efficiency of knockdown should be shown even if worms eat less food.
- 6) Intron retention is only one of at least 5 defined alternative splicing types. The authors should be careful claiming DR increases AS in general, but rather focus on the only type of AS they have included in their analyses. The paragraph titles should reflect their method of DR as others have reported on AS phenomenon already using other DR protocols. The authors should further acknowledge the work of Tabrez et al 2017 in their results section, who

already coupled nonsense mediated decay to DR mediated longevity in the last chapter of the results section. There is a discrepancy between the Tabrez and the current study about which smg genes are required for DR longevity. The authors should comment on this finding. Do the authors have any evidence on mRNA level for the effects of smg-6 and smg-7 mutants on transcript regulation in their study? What are the levels of smg genes in their gene expression data?

7) A final schematic would be helpful to summarize the different regulatory processes assessed that contribute to post-transcriptional regulation by DR.

Minor comments:

- 1) -Experimental details for the worm culture for the RNA-seq and polysome profiling in addition to worm strain information are missing.
- 2) Figure 5 legend: it's not long-rank test
- 3) Genes for respective RNA binding proteins are *tiar-1*, *tiar-2* and *tiar-3* and not *tair-* as written in the main text
- 4) The recent paper by Rhoads et al 2018 on CR in monkeys should be discussed in the alternative splicing section discussion
- 5) The different shades of green and red in Figure 2E are difficult to distinguish.
- 6) In the RNA editing paragraph, reference to figure EV4A is probably wrong
- 7) Is the *trxr-1* splicing event considered rather an alternative splice site usage event than intron retention?

Reviewer #3 (Comments to the Authors (Required)):

This is a well-written paper that delves into the relationship between the transcriptomes of total and polysomal RNAs in worms subject to dietary restriction. In general, it was an interesting and enjoyable paper to read and the topic is very suitable for the readership of LSA. There are a few points that should be addressed before publication.

Major points:

- 1) The authors use DPAR calculation to identify genes that exhibit a net translation change. The observation that the vast majority of translational regulated genes do not show a transcriptional change is quite striking. Would it be possible to confirm by western blot (antibody availability permitting) some of the top hits?
- 2) Where are the lifespan curves for the *unc-75* and *asd-2* knockdowns? The paragraph refers to no figures or tables and I don't see any lifespan curves in the main body of the paper or in the Supp using these knockdowns. I don't think with these assays that listing the percentage of lifespan increase is enough, especially if the authors wish to use the bold phrase, "...to our knowledge, this is the first study to show *unc-75* is a robust regulator of longevity." (In general, I would discourage priority claims such as with the word "first").
- 3) The M&M needs some work. The methods of dietary restriction by dilution should be clearly stated as well as how the RNAi was done and what sequences were used to generate the knockdown constructs. At the moment this info seems to be missing.
- 4) Judging from table EV6 it looks as if the RNAi lifespans were performed in duplicate. The requirement these days with RNAi lifespans especially (where the knockdown efficiencies may vary)

is that they be performed in triplicate (and the number of replicates should be mentioned in the M&M).

5) It's not clear how the RNAi experiments in Figure 5C can be performed in tandem with the dilution method of DR. An equal amount of bacteria expressing the RNAi construct needs to be available to all the worms, if the RNAi-expressing bacteria is diluted in DR-treated worms then these worms will not likely knock down the gene efficiently. A qPCR demonstrating equally efficient knockdowns of *mex-5*, *pab-1*, and *sup-26* in AL and DR conditions are necessary here, and it should be made clear in the M&M how these experiments are designed to get around the dilution issue.

6) In the fourth paragraph under "Annotated RNA-binding sequence motifs..." where the authors comment the lifespan on RNAi results the text and the graph labels do not match (*sup-26* is on the right and *pab-1* in the middle). Moreover, in the most right graph the RNAi treatment knocking down *sup-26* does seem to increase the lifespan of DR worms compared L4440 DR worms, but the text states otherwise. Are the authors possibly referring to a different curve or is this merely a non-significant increase? It looks like a significant increase to me, although the data in the EV6 table states otherwise, so this needs to be cleared up.

7) An interesting part of the study is in the A to I RNA editing section. I would like to see a table that includes the genes with putative editing sites and the GO analysis of these genes would be nice. Is there a reduction in the translation efficiency of any genes that might contribute to the lifespan extension of the DR worms?

8) I think in order for the authors to convincingly demonstrate that the reduction A to I editing occurring on *eif2-alpha* is a direct result of ADAR activity that they need to show that DR reduces the mRNA and/or protein levels (or enzymatic activity if possible) of the ADARs.

9) It is puzzling that the expression of *eif-2alpha* is up in both DR_TO and DR_TR compared to AL_TO and AL_TR given that the literature suggests that 1) the translational efficiency of many genes that positively regulate translation are reduced under conditions that extend lifespan in worms 2) global translation rates are down as is the protein level of both *eif-1* and *eif-3* upon caloric restriction in mutants like *eat-2* (Yuan et al JBC 2012). How do the authors explain then that DR, which should reduce caloric intake and slow the mTOR pathway, results in an increase in the expression of *eif-2alpha*? This needs to be addressed.

10) I know this may be an impossible task, but is there an antibody available that can determine if the editing of the *eif-2alpha* mRNA (or any other edited mRNA) changes the protein level? Or, are the authors able to provide a Sashimi plot from their RNA-Seq data showing the editing at these sites introduces or removes a splice site? As the data stands now it is difficult to determine if the editing of the mRNA is biologically relevant or artifactual.

11) The authors need to show that *mir-58* and *mir-80* levels are also downregulated specifically in their experiments in order for Figure 5D to be convincing. Referring to other experiments in other labs that show this knockdown doesn't cut it.

Minor points:

1) I don't think the authors can make the statement "...the robust translational promotion of *eif2-alpha* under AL conditions was partially attenuated under DR (DPAR of -0.295, Figure EV3D)"

without proving some sort of calculation.

2) A more pictorial representation of what is going in the eif2-alpha gene would be helpful (e.g. a gene structure with the 3'UTR editing sites clearly delineated). Is anything known about these specific sequences in the 3'UTR? Such as, do the edits result in a gain or loss of microRNA binding sites or RBP sites?

3) With the EV1 table I repeated the DAVID analysis with the latest 6.8 version (the authors use 6.7). One potential issue is the failure of many gene to convert to Entrez. For example, of the 257 up-regulated genes under DR only 156 converted successfully and only 533 of the 782 down-regulated genes converted. If the authors found similar incomplete conversion rates, please state this in the manuscript and allow for the possibility that the GO analysis may change upon annotation of the genes that fail to convert. Also, with the 6.8 version the significance of the enriched terms changes, for example I found that the top most significantly enriched terms for the down-regulated mRNAs under DR relate to immunity and defense and not apoptosis. The authors may wish to re-run their analyses using the more recent version.

4) There should be some citations mentioned to make the following statement about pab-1, "However, it is primarily considered a pro-translation factor."

5) The labeling of Table EV6 should be improved, the way it is presented now with the labels running into each other is confusing. Putting the data into a more formatted table would help.

6) Is there a reason why sup-12, etr-1 and sap-49 are left outside of the lifespan analysis in Figure 5 despite having a similar enrichment profile? If so, it should be stated.

Reviewer #2 (Comments to the Authors (Required)):

Rollins et al. report here an extensive analysis on the transcriptional and translational regulation induced by bacterial food dilution in *C. elegans*. The differentiation between transcriptome and translational regulation with analysis in the same samples appears novel to the DR field. The authors correlate gene expression changes with the abundance of mRNAs in the polysome-associated mRNA fractions. As a consequence, they establish a list of mRNAs with translational suppression or promotion by DR. In their further study they highlight a few novel DR longevity genes and investigate functional analysis. To add mechanistic details for the suppression or promotion of translation in DR, they in silico investigate mRNA processing events affecting translatability such as location of genes in operons, trans-splicing, RNA editing, intron retention and nonsense-mediated decay of their previously selected genes with translational regulation for involvement in promoting DR-mediated longevity.

This manuscript adds to the understanding of the mechanisms by which DR might promote healthier aging. Transcriptome analysis of various DR protocols has been reported multiple times, but most published studies were only focused on gene expression changes. This work focuses on the mRNAs actually translated, adds different aspects of post-transcriptional regulation to the mechanism and constitutes a valuable resource to the community about the effects of DR in *C. elegans*.

However, the manuscript is focused on in silico analysis only, mostly descriptive and correlative, with little experimental laboratory validation to support the computational analysis and claims are often of speculative nature. Literature is appropriately discussed; although at times more recent studies should be included in the discussion. For this reviewer, the missing experimental validation and the limited technical details reduce the excitement for this study. Major comments and questions:

****1) What is the reasoning behind comparing the data to IF instead/in addition to comparing to other DR transcriptome analyses? How do the gene expression results compare to other DR transcriptome analyses?***

The rationale of the comparison was to be able to show that our mRNA-seq pipeline was able to recapitulate published transcriptomic responses to caloric restriction in *C. elegans*. Despite the fact that many transcriptomic studies have been performed in worms related to changes in dietary factors, there is a surprising paucity of information on types of dietary restriction invoked during adulthood in the manner carried out in this study. One-to-one comparison to other organisms under DR is certainly possible but would not serve as a good point of reference due to issues in predicting homology and assuming that similar genes always have similar functions. Instead, we compared the biological functions GO terms enriched in our transcriptomic analysis of DR to a previous meta-analysis across several species to provide insight into what aspects of DR are evolutionary conserved and which are different.

****2) Since DR slows global translation rates, how does this affect polysome profiling at a defined time point and comparison to samples with a different translation rate? Does this impact the DPAR? Different translation rates should be discussed.***

Typical mRNAseq pipelines quantify the abundance of individual transcripts within a sample relative to the total number transcripts present in that sample. Thus, of those present at a given timepoint, we can determine the proportion belonging to a particular gene. Quantifying transcript abundance among polysomal mRNA is no different. We added the following text on page 7 to clarify that the DPAR calculation is not an absolute measurement: " It should be noted that, while global translation rates are lower under DR, DPAR is only influenced by relative changes in the distribution of specific mRNAs within polysomes and total mRNA.."

***3) The number of quantified annotated coding genes is rather low. Is there a technical explanation?**

For a coding gene to be considered as "quantified" or "quantifiable", the number of average reads must be 8 or more per million reads averaged across the entire dataset. As described in the material and methods, this threshold was chosen as it was the point where the trend in the biological coefficient of variation approached the common variation of the entire dataset as estimated from a dispersion plot (shown below). As this manuscript relied heavily on *in silico* analysis we wanted to reduce the number of false positives that would arise from considering genes with high variation. The trade-off of this threshold is that very lowly expressed genes were necessarily excluded from the downstream analysis.

***4) What is the number of replicates for the lifespan results in Table 1? Some of the lifespan increases are less than 10% and 3 replicates should be presented for statistical relevance to support the claim for these genes and their potential role in DR longevity.**

Further, what is the reasoning for using FUDR in fully fed conditions? The use of FUDR is controversial in the *C. elegans* field as it has been shown to prolong lifespan by itself and the authors should show the lifespan results without the use of FUDR.

The lifespans in Table 1 were based on two biological replicates that were analyzed together. To address reproducibility, we conducted two more additional replicates of this assay. One was assay was performed with FUDR and the other without. The assay without FUDR gave similar results to those

conducted with it. Instead of Table 1, we have included the lifespans curves from one of the biological replicates as Figure 2G. The median lifespan, percent change, and p-value for each of the biological replicates analyzed separately are now included in supplemental Figure S6.

Did the authors validate gene expression levels of the novel DR longevity regulators by alternative methods?

We did not. Systematic validation of expression of the novel longevity regulators was not viable due to the paucity of antibodies available for use in *C. elegans*.

****Have the authors assessed the efficiency of the RNAi knockdown?***

The gene expression levels of the longevity screen hits were not quantified. The RNAi clones that gave positive hits from the initial screen were sequenced validated and previously published effects on development were verified prior to the lifespan assays.

****Lifespan increase for several genes in Table 1 is much less than the longevity seen with DR itself. In order to really prove that these are novel longevity regulators, I would suggest to perform the converse experiment and increase the concentration of respective genes in a DR background and test if DR then fails to extend lifespan.***

We agree that the question remains whether decreased expression is required for DR induced longevity. However, increasing gene expression in *C. elegans* is not a trivial task, especially as compared to knockdowns by RNAi. These genes form the basis of ongoing DR gene function studies. As we continue analysis, we note an important consideration: overexpression of a gene in *C. elegans* often involves the formation of extra-chromosomal arrays which can vary greatly in copy number and expression. There is no guarantee that such transgenics will not be embryonic lethal, shorter lived, or have sickly phenotypes that will confound the effects of DR. To more succinctly highlight the distinction made by the reviewer, we changed the title of the section involving these lifespan assays to “RNAi screen of translationally regulated genes under DR revealed novel longevity genes”.

****5) How does food dilution affect RNAi knockdown in lifespans for RNA binding proteins? Do you achieve sufficient knockdown with reduced food intake? Efficiency of knockdown should be shown even if worms eat less food.***

Lifespans of the RNA binding proteins were performed by first inducing RNAi for two days using fully fed (AL) bacterial lawn densities and then transferring to AL or DR conditions with OP50. We have previously shown (Howard et al. Aging Cell, 2016) that RNAi can reduce protein expression by 70% in two days. Due to endogenous RNAi amplification, knockdown persists long after initial exposure to the dsRNA. Unfortunately, quantification of knockdown is frequently obstructed by continued RNAi templating by endogenous machinery, which can effect qPCR even when primers are designed outside the region of dsRNA used to induce RNAi. The materials and methods on page 25 have been updated to clarify the procedure used.

****6) Intron retention is only one of at least 5 defined alternative splicing types. The authors should be***

careful claiming DR increases AS in general, but rather focus on the only type of AS they have included in their analyses. The paragraph titles should reflect their method of DR as others have reported on AS phenomenon already using other DR protocols.

The text on page 20 has been updated to reflect that only intron retention was considered in the analysis. Additionally, the text on page 14 has been updated to specify that it is DR by bacterial dilution.

****The authors should further acknowledge the work of Tabrez et al 2017 in their results section, who already coupled nonsense mediated decay to DR mediated longevity in the last chapter of the results section. There is a discrepancy between the Tabrez and the current study about which smg genes are required for DR longevity. The authors should comment on this finding.***

We now state on page 16 of the results that “The requirement of *smg-2* for the full effects of DR induced longevity in the *eat-2* mutant and by bacterial dilution was previously shown (Tabrez et al, 2017)”. We see no direct discrepancy between our results and theirs about the role of *smg* genes in longevity. Our lifespans using the *smg-2* mutant also showed that the effects of DR on longevity were muted as did the Tabrez study on bacterial dilution. However, Tabrez et al did not check the effects of DR in the *smg-6* and *smg-7* mutants as in the current study.

****Do the authors have any evidence on mRNA level for the effects of smg-6 and smg-7 mutants on transcript regulation in their study?***

We have performed polysome profiling on the *smg-6* and *smg-7* mutants and have quantified the expression of NMD prone transcripts in these mutants. However, these data are being used in a follow-up manuscript in which we elucidate the mechanism of the effects of these mutants on longevity.

****What are the levels of smg genes in their gene expression data?***

A bar chart of the expression of *smg-1/2/3/4/5/6/7* in the DR dataset has been provided in figure EV3D.

****7) A final schematic would be helpful to summarize the different regulatory processes assessed that contribute to post-transcriptional regulation by DR.***

A schematic has been added as supplemental Figure S5 depicting the regulatory processes affecting selective translation considered in this manuscript.

Minor comments:

****1) -Experimental details for the worm culture for the RNA-seq and polysome profiling in addition to worm strain information are missing.***

The procedure used for worm culture for the RNA-seq experiment has been included in the material and methods on page 22. The worm strains used have been added on page 25.

****2) Figure 5 legend: it's not long-rank test .***

The text has been corrected to log-rank text.

***3) Genes for respective RNA binding proteins are *tiar-1*, *tiar-2* and *tiar-3* and not *tair-* as written in the main text**

This typo has been corrected.

***4) The recent paper by Rhoads et al 2018 on CR in monkeys should be discussed in the alternative splicing section discussion.**

We have included the following reference to the CR primate study as evidence of the link between CR and alternative splicing: “Furthermore, hepatic transcripts in caloric restricted primates exhibited increased alternative exon use, including genes related to fatty acid metabolism. (Rhoads *et al*, 2018)”

***5) The different shades of green and red in Figure 2E are difficult to distinguish.**

The shades have been optimized to enhance the contrast of figure 2E.

***6) In the RNA editing paragraph, reference to figure EV4A is probably wrong**

This typo has been corrected to correctly state *EV3A*.

***7) Is the *trxr-1* splicing event considered rather an alternative splice site usage event than intron retention?**

Admittedly, we struggled with this nomenclature as well. This question brings up the difference between partial intron retention and alternative acceptor/donor site usage. Both are used in the alternative splicing field interchangeably. As the isoform of *trxr-1* detected in our mRNA-seq analysis included a portion of the transcript that was previously annotated as an intron, we chose to define the event as partial intron retention for simplicity and clarity.

Reviewer #3 (Comments to the Authors (Required)):

This is a well-written paper that delves into the relationship between the transcriptomes of total and polysomal RNAs in worms subject to dietary restriction. In general, it was an interesting and enjoyable paper to read and the topic is very suitable for the readership of LSA. There are a few points that should be addressed before publication.

Major points:

1) The authors use DPAR calculation to identify genes that exhibit a net translation change. The observation that the vast majority of translational regulated genes do not show a transcriptional change is quite striking. Would it be possible to confirm by western blot (antibody availability permitting) some of the top hits?

Unfortunately, due to the paucity of antibodies available in *C. elegans* it was not feasible to systemically confirm regulation on the protein level by western blot.

***2) Where are the lifespan curves for the *unc-75* and *asd-2* knockdowns? The paragraph refers to no figures or tables and I don't see any lifespan curves in the main body of the paper or in the Supp using these knockdowns. I don't think with these assays that listing the percentage of lifespan increase is enough, especially if the authors wish to use the bold phrase, "...to our knowledge, this is the first study to show *unc-75* is a robust regulator of longevity." (In general, I would discourage priority claims such as with the word "first").**

Representative lifespans for *unc-75* and *asd-2* are now included in figure 5, instead of only in the supplemental table S8. The text on page 13 is changed to state that *unc-75* RNAi was not previously annotated as having a longevity effect.

***3) The M&M needs some work. The methods of dietary restriction by dilution should be clearly stated as well as how the RNAi was done and what sequences were used to generate the knockdown constructs. At the moment this info seems to be missing.**

This is now addressed in the section 'RNA interference by feeding and dietary restriction' and has been added to the material and methods on page 25.

***4) Judging from table EV6 it looks as if the RNAi lifespans were performed in duplicate. The requirement these days with RNAi lifespans especially (where the knockdown efficiencies may vary) is that they be performed in triplicate (and the number of replicates should be mentioned in the M&M).**

As recommended, we have performed a third biological replicate of these lifespans and they have been included in Table S6. Additionally, the materials and methods have been revised to state that 3 replicates were used.

***5) It's not clear how the RNAi experiments in Figure 5C can be performed in tandem with the dilution method of DR. An equal amount of bacteria expressing the RNAi construct needs to be available to all the worms, if the RNAi-expressing bacteria is diluted in DR-treated worms then these worms will not likely knock down the gene efficiently. A qPCR demonstrating equally efficient knockdowns of *mex-5*, *pab-1*, and *sup-26* in AL and DR conditions are necessary here, and it should be made clear in the M&M how these experiments are designed to get around the dilution issue.**

Lifespans of the RNA binding proteins were performed by first inducing RNAi for two days using fully fed (AL) bacterial lawn densities and then transferring to AL or DR conditions with OP50. Due to endogenous RNAi amplification, knockdown persists long after initial exposure to the dsRNA. We have previously shown in (Howard et al 2014), that high levels of knockdown can be achieved on the protein level after two days of RNAi feeding. The section "RNA interference by feeding" has been added to the materials and methods to clarify the procedure used.

***6) In the fourth paragraph under "Annotated RNA-binding sequence motifs..." where the authors**

comment the lifespan on RNAi results the text and the graph labels do not match (sup-26 is on the right and pab-1 in the middle). Moreover, in the most right graph the RNAi treatment knocking down sup-26 does seem to increase the lifespan of DR worms compared L4440 DR worms, but the text states otherwise. Are the authors possibly referring to a different curve or is this “merely a non-significant increase? It looks like a significant increase to me, although the data in the EV6 table states otherwise, so this needs to be cleared up.

The error in the main text has been corrected to match the (correctly labeled) graph. The perceived increase in the representative lifespan due to *sup-26* RNAi under DR was close to statistical significance ($p=0.054729$), but didn't pass the commonly used threshold of $p < 0.05$.

****7) An interesting part of the study is in the A to I RNA editing section. I would like to see a table that includes the genes with putative editing sites and the GO analysis of these genes would be nice. Is there a reduction in the translation efficiency of any genes that might contribute to the lifespan extension of the DR worms?***

We have included a new supplemental table S7 of genes with editing in at least one group. The table includes the position of putative editing sites, their mean editing frequency in each of the treatments and fractions and the gene associated with the edit site. In the same supplemental, we have also included the results of the GO term enrichment for biological processes of these genes. Some of the enriched terms include 'proteasomal protein catabolic process', 'cellular protein modification process', and 'cell cycle'. Text on page 11 has been updated to reflect this additional data. Prior to submission, we looked extensively for examples of longevity genes that were edited under DR and have altered translation rates but found none.

****8) I think in order for the authors to convincingly demonstrate that the reduction A to I editing occurring on eif2-alpha is a direct result of ADAR activity that they need to show that DR reduces the mRNA and/or protein levels (or enzymatic activity if possible) of the ADARs.***

It was previously shown that some of the same editing sites in *eif2-alpha* we report were also changed upon deletion of *adr-1* in *C. elegans* which is highly suggestive that they resulted by its activity. Additionally, we have included the plotted expression levels of *adr-1*, *adr-2* and *adbp-1* under DR from our dataset showing that *adr-1* is downregulated under DR (Table EV3D). We have added the following text on page 12: "The expression of the A-to-I editing regulator gene *adr-1* and the gene encoding its binding partner *adbp-1* was reduced in the translated fraction under DR (Figure S3D). Conversely, the expression of *adr-2* was slightly increased under DR. These expression changes in ADAR are a likely explanation of the changes in editing frequencies observed under DR.

****9) It is puzzling that the expression of eif-2alpha is up in both DR_TO and DR_TR compared to AL_TO and AL_TR given that the literature suggests that 1) the translational efficiency of many genes that positively regulate translation are reduced under conditions that extend lifespan in worms 2) global translation rates are down as is the protein level of both eif-1 and eif-3 upon caloric restriction in mutants like eat-2 (Yuan et al JBC 2012). How do the authors explain then that DR, which should reduce caloric intake and slow the mTOR pathway, results in an increase in the expression of eif-2alpha? This needs to be addressed.***

Although it is up under DR_TR, this is merely a reflection of its relative increase in transcript level, which under zero translational change, would be reflected in the higher DR_TR observed. Because the DR_TR didn't increase as much as the transcript expression, there is a small decrease in translational efficiency. This is not at all an uncommon phenomenon in our experience. That said, we do not have a good explanation for the increase in eif-2alpha transcript expression. However, the ability of eif-2alpha to regulate translation rates is post-translational and generally independent of TOR. Instead, its activity is regulated by kinases involved in ER_UPR (PERK/PEK-1) and downstream of uncharged tRNA sensing (GCN2/GCN-2). Phosphorylated EIF-2ALPHA then inhibits replenishment of the ternary complex and lowers translation. It was previously shown (Rousakis et al. Aging Cell, 2013) that this phosphorylation is increased in *C. elegans* in response to an increase in uncharged tRNAs, as expected from studies in other organisms. Thus, it appears that the post-translational regulation of EIF-2ALPHA is important for responses to DR.

****10) I know this may be an impossible task, but is there an antibody available that can determine if the editing of the eif-2alpha mRNA (or any other edited mRNA) changes the protein level? Or, are the authors able to provide a Sashimi plot from their RNA-Seq data showing the editing at these sites introduces or removes a splice site? As the data stands now it is difficult to determine if the editing of the mRNA is biologically relevant or artifactual.***

Unfortunately, antibodies specific to EIF-2ALPHA is not commercially available in *C. elegans* to test the effect of the edits on the protein level. There was no evidence from the mRNA-seq that edit containing reads aligning to *eif2-alpha* were being alternatively spliced, so the edits don't appear to introduce or remove a splice site. It is possible that the edit at 2076770 alters interactions with miRNA as it occurs in the binding sites of mir-792 and mir-1820. The edit site 2076705 may mediate interactions with RNABPs as it occurs at within a stemloop, which is a common site for RNA-protein interactions. The predicted secondary structure of eif-2alpha has been added to Figure S3 F. The possible role of stem loop binding proteins and miRNA in the regulation of *eif2-alpha* have been added on page 12.

****11) The authors need to show that mir-58 and mir-80 levels are also downregulated specifically in their experiments in order for Figure 5D to be convincing. Referring to other experiments in other labs that show this knockdown doesn't cut it.***

Using qPCR, we quantified the expression of all microRNA families that were implicated by the in silico analysis under AL and DR conditions. This analysis showed a significant down regulation of mir-58a-3p, mir-80-3p, mir-81-3p and mir-82-3p (Figure 6B). However, no other of the DPAR enriched miRNA were significantly altered under DR. This result is now described on page 14 and the experiment added to the materials and methods.

Minor points:

****1) I don't think the authors can make the statement "...the robust translational promotion of eif2-alpha under AL conditions was partially attenuated under DR (DPAR of -0.295, Figure EV3D)" without proving some sort of calculation.***

The DPAR calculation is explained in the Materials and Methods section under Differential Expression (Page 22).

***2) A more pictorial representation of what is going in the eif2-alpha gene would be helpful (e.g. a gene structure with the 3'UTR editing sites clearly delineated). Is anything known about these specific sequences in the 3'UTR? Such as, do the edits result in a gain or loss of microRNA binding sites or RBP sites?**

This point has been addressed as part of our response to major point #11 above.

***3) With the EV1 table I repeated the DAVID analysis with the latest 6.8 version (the authors use 6.7). One potential issue is the failure of many gene to convert to Entrez. For example, of the 257 up-regulated genes under DR only 156 converted successfully and only 533 of the 782 down-regulated genes converted. If the authors found similar incomplete conversion rates, please state this in the manuscript and allow for the possibility that the GO analysis may change upon annotation of the genes that fail to convert. Also, with the 6.8 version the significance of the enriched terms changes, for example I found that the top most significantly enriched terms for the down-regulated mRNAs under DR relate to immunity and defense and not apoptosis. The authors may wish to re-run their analyses using the more recent version.**

We have also seen a continuous change in enriched GO terms as versions change and annotations are added over the years. More specifically for *C. elegans* research, we have also experienced low conversion rates from Wormbase IDS to Entrez similar to what the reviewer experienced. The number of genes successfully converted for each GO analysis is provided in each table as 'List.Total' and 'Pop.Total'. However, to address this possible explicitly, we have added the following text to the materials and methods on page : "Genes lists submitted to DAVID analysis often had low conversion rates (<80%), therefore it is likely that enriched terms will change as annotations become more complete."

***4) There should be some citations mentioned to make the following statement about pab-1, "However, it is primarily considered a pro-translation factor."**

Two citations have been added which review the roles of *pab-1*.

***5) The labeling of Table EV6 should be improved, the way it is presented now with the labels running into each other is confusing. Putting the data into a more formatted table would help.**

There was an error in the conversion process. The column widths are now optimized so that labels do not overlap.

***6) Is there a reason why sup-12, etr-1 and sap-49 are left outside of the lifespan analysis in Figure 5 despite having a similar enrichment profile? If so, it should be stated.**

The RNAi for these genes were not checked for an effect on lifespan as they were either not available in our library or because we were not able to sequence validate them.

June 18, 2019

RE: Life Science Alliance Manuscript #LSA-2018-00281-TR

Dr. Jarod A Rollins
MDI Biological Laboratory
159 Old Bar Harbor Road
Bar Harbor, Maine 04672

Dear Dr. Rollins,

Thank you for submitting your revised manuscript entitled "Dietary restriction induces post-transcriptional regulation of longevity genes". One of the original reviewers re-assessed your work. The reviewer appreciates the introduced changes and we would thus be happy to publish your paper in Life Science Alliance pending final revisions necessary to meet our formatting guidelines:

- please add a callout in the manuscript text to Fig7E, Suppl Fig 4B and D.
- please note that you mention Table EV10 in the legend of Fig 7 => please change to Table S10.

A. FINAL FILES:

B. MANUSCRIPT ORGANIZATION AND FORMATTING:

Sincerely,

Reviewer #3 (Comments to the Authors (Required)):

This is a revision of work submitted by Rollins et al. In the revision the authors have duly responded to the concerns of this reviewer, the additional experiments requested have been added except in

the cases where the lack of available reagents renders the request not possible. I have no further comments and support the publication of this work in LSA.

June 20, 2019

RE: Life Science Alliance Manuscript #LSA-2018-00281-TRR

Dr. Jarod A Rollins
MDI Biological Laboratory
159 Old Bar Harbor Road
Bar Harbor, Maine 04672

Dear Dr. Rollins,

Thank you for submitting your Research Article entitled "Dietary restriction induces post-transcriptional regulation of longevity genes.". It is a pleasure to let you know that your manuscript is now accepted for publication in Life Science Alliance. Congratulations on this interesting work.

DISTRIBUTION OF MATERIALS:

Again, congratulations on a very nice paper. I hope you found the review process to be constructive and are pleased with how the manuscript was handled editorially. We look forward to future exciting submissions from your lab.

Sincerely,
